# AGENTIC-KGR: CO-EVOLUTIONARY KNOWLEDGE GRAPH CONSTRUCTION THROUGH MULTI-AGENT REINFORCEMENT LEARNING

## ABSTRACT

Current knowledge-enhanced large language models (LLMs) rely on static, pre-constructed knowledge bases that suffer from coverage gaps and temporal obsolescence, limiting their effectiveness in dynamic information environments. We present Agentic-KGR, a novel framework enabling co-evolution between LLMs and knowledge graphs (KGs) through multi-round reinforcement learning (RL). Our approach introduces three key innovations: (1) a dynamic schema expansion mechanism that systematically extends graph ontologies beyond pre-defined boundaries during training; (2) a retrieval-augmented memory system enabling synergistic co-evolution between model parameters and knowledge structures through continuous optimization; (3) a learnable multi-scale prompt compression approach that preserves critical information while reducing computational complexity through adaptive sequence optimization. Experimental results demonstrate substantial improvements over supervised baselines and single-round RL approaches in knowledge extraction tasks. When integrated with GraphRAG, our method achieves superior performance in downstream QA tasks, with significant gains in both accuracy and knowledge coverage compared to existing methods.

## 1 INTRODUCTION

Large Language Models (LLMs) have revolutionized natural language processing and knowledge-intensive applications, demonstrating remarkable capabilities in understanding and generating human-like text. However, their susceptibility to hallucination and limited access to up-to-date information pose significant challenges for reliable knowledge-based reasoning tasks. Knowledge graphs (KGs), with their structured representation of entities and relationships, offer a promising solution to enhance LLM reliability by providing factual grounding (Nie et al., 2023). The integration of LLMs with KGs has emerged as a critical research direction, particularly in developing intelligent QA systems that require both comprehensive knowledge coverage and accurate reasoning capabilities. Recent advances in Graph Retrieval-Augmented Generation (GraphRAG) have shown substantial improvements in mitigating hallucination by incorporating structured knowledge into the generation process (Luo et al., 2025; Lelong et al., 2025).

The landscape of KG reasoning and RAG has witnessed significant methodological advances in recent years. Reinforcement learning (RL) approaches have proven particularly effective for multi-hop reasoning over incomplete KGs, with pioneering works like DeepPath introducing policy-based agents that learn to navigate KG vector spaces by sampling promising relational paths (Xiong et al., 2018). Building upon this foundation, MINERVA addressed the more challenging task of QA with known relations but single entities, employing neural RL to navigate graphs conditioned on input queries (Das et al., 2018). To address the issue of low-quality rewards in incomplete KG environments, researchers have developed sophisticated reward shaping mechanisms and self-supervised pre-training methods (Lin et al., 2018; Ma et al., 2025). Recent developments have focused on agentic RAG systems that employ multi-tool architectures for iterative, targeted queries and multi-hop reasoning (Lelong et al., 2025). The emergence of Graph-R1 has introduced lightweight knowledge hypergraph construction with multi-turn agent-environment interactions optimized through end-to-end reward mechanisms (Luo et al., 2025). Furthermore, advances in KG construction have leveraged LLMs as automatic constructors, with frameworks like SAC-KG demonstrating the potential for automated domain-specific KG generation (Chen et al., 2024).

Despite these advances, current GraphRAG approaches face critical limitations that constrain their real-world applicability. Existing methods rely on static, pre-constructed KGs that suffer from coverage gaps, temporal obsolescence, and inability to adapt to emerging domain knowledge or evolving query patterns. Traditional RL approaches for knowledge reasoning focus primarily on path-finding within fixed graph structures, neglecting the potential for co-evolution

between reasoning agents and knowledge bases. Furthermore, the separation between knowledge construction and utilization creates a fundamental bottleneck where sophisticated reasoning mechanisms remain constrained by static knowledge repositories. Current single-objective optimization strategies fail to balance the dual requirements of effective knowledge extraction and accurate question answering, resulting in suboptimal integrated system performance. These limitations necessitate a paradigm shift toward adaptive knowledge systems that can dynamically construct, expand, and refine KGs through iterative interaction with both data sources and reasoning tasks.

To address these fundamental limitations, we propose Agentic-KGR, a novel framework that enables co-evolution between LLMs and KGs through multi-round RL. The core innovation lies in reimagining knowledge construction and utilization as interconnected, mutually reinforcing processes rather than sequential stages, as demonstrated in typical product QA scenario (Figure 1). Our framework introduces three key contributions:

- a dynamic ontological expansion framework that facilitates real-time structural evolution of knowledge graphs through adaptive schema augmentation;

- a co-evolutionary memory architecture that enables bidirectional adaptation between neural representations and knowledge structures through iterative refinement processes;

- a learnable multi-scale prompt compressor with cross-attention mechanisms that achieves backbone-agnostic semantic preservation while reducing computational overhead through adaptive query-based context distillation.

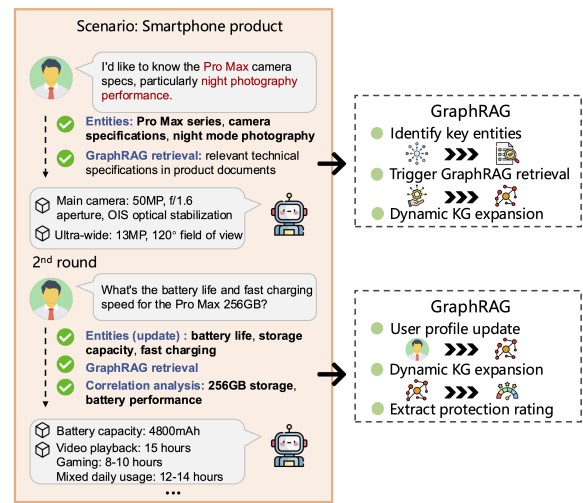

Figure 1: Multi-round interactive knowledge discovery in product QA scenario

This multi-round RL paradigm ensures synergistic evolution of both the reasoning agent and knowledge structure, where improvements in one component enhance overall system performance, fundamentally transforming the paradigm from static knowledge retrieval to adaptive knowledge co-creation.

Experimental evaluation demonstrates the effectiveness of Agentic-KGR across two critical dimensions: KG extraction and QA performance. In knowledge extraction tasks, models trained with our framework exhibit substantial improvements in graph density, coverage, and quality compared to supervised fine-tuning baselines and single-round RL approaches. The dual reward mechanism proves particularly effective in balancing exploration of novel knowledge territories with exploitation of established patterns, enabling the extraction of more comprehensive and accurate KGs. Specifically, Agentic-KGR achieves up to +33.3 points improvement over existing RL methods in graph extraction and +12.8 points in downstream QA tasks, demonstrating that multi-round agentic RL effectively enhances both KG construction quality and end-to-end task performance through improved coverage and evidence routing. When these dynamically constructed KGs are integrated into GraphRAG systems, they significantly enhance downstream QA performance, validating the synergistic relationship between improved knowledge extraction and reasoning capabilities. This two-stage validation confirms that Agentic-KGR's ability to generate high-quality, domain-adaptive KGs establishes a foundation for self-improving knowledge systems that can continuously evolve alongside their operational environments.

## 2 METHOD

Figure 2 illustrates the overall architecture of our Agentic-KGR framework, which integrates a comprehensive tool pool for knowledge graph operations with a dual reward mechanism, allowing the model to dynamically construct and expand knowledge graphs while simultaneously improving its reasoning capabilities.

### 2.1 PRELIMINARIES

**Notation.** We denote vectors by lowercase bold letters (e.g., $\mathbf{x}, \mathbf{h}$), and matrices by capital bold letters (e.g., $\mathbf{A}, \mathbf{H}$). Let $\| \cdot \|_2, \| \cdot \|_F$ be the $\ell_2$-norm and Frobenius norm respectively. We use $\langle \cdot, \cdot \rangle$ for inner products, $\nabla$ for gradients, and $\mathrm{tr}(\cdot)$ for matrix trace. Let a LLM be parameterized by $\theta \in \mathbb{R}^p$. The KG at round $t$ is $\mathcal{G}_t \equiv (V_t, E_t, \ell_t)$ with

Figure 2: Overall Architecture of Agentic-KGR Framework.

Laplacian $\mathbf{L}_t$. We model the process as a POMDP with latent state $x_t \in \mathcal{X}$, observation $s_t \in \mathcal{S}$, action $a_t \in \mathcal{A}$, query $q_t \in \mathcal{Q}$, external memory $\mathcal{M}_t$, and policy $\pi_\theta$. The dual reward is $R_t = \alpha R_{\text{env}}(s_t, a_t) + (1 - \alpha)R_{\text{task}}(s_t, a_t)$ with $\alpha \in [0, 1]$ and discount $\gamma \in (0, 1)$. For compression, $\mathbf{H} \in \mathbb{R}^{n \times d}$ is the uncompressed embedding matrix, $\mathbf{Z} \in \mathbb{R}^{k \times d}$ is the compressed representation with $k \ll n$. We use $\phi(\cdot)$ for compression, $\psi(\cdot)$ for reconstruction, $I(\cdot; \cdot)$ for mutual information, and $\text{KL}(\cdot\|\cdot)$ for KL-divergence.

**Definitions.** We present the following definitions that formalize the core components of our Agentic-KGR framework.

**Definition 1** (Differentiable Subgraph Retrieval Distribution). Given query $q_t \in \mathcal{Q}$ and knowledge graph $\mathcal{G}_t$, we define a parametric probability measure $\mu_\eta$ over the power set $2^{E_t}$ of edge subsets. The differentiable retrieval distribution over subgraphs $\mathcal{H}_t \subseteq \mathcal{G}_t$ is characterized by the Gibbs measure:

$$\mathbb{P}_\eta(\mathcal{H}_t \mid q_t, \mathcal{G}_t) = \frac{1}{Z_\eta(q_t, \mathcal{G}_t)} \exp\left(\beta \cdot \text{score}_\eta(q_t, \mathcal{H}_t)\right), \tag{1}$$

where $\hat{Z}_\eta(q_t, \mathcal{G}_t) = \frac{1}{M}\sum_{m=1}^{M} \exp(\beta \cdot \text{score}_\eta(q_t, \mathcal{H}_m))$ is the partition function, and the energy function combines semantic alignment with spectral coherence:

$$\text{score}_\eta(q_t, \mathcal{H}_t) = \langle \phi_\eta(q_t), \Phi_\eta(\mathcal{H}_t) \rangle - \lambda_{\text{spec}}\left[\text{tr}(\mathbf{L}_{\mathcal{H}_t}^+) + \frac{1}{2}\log\det(\mathbf{L}_{\mathcal{H}_t} + \epsilon\mathbf{I})\right], \tag{2}$$

with temperature parameter $\beta > 0$, regularization $\epsilon > 0$, and $\mathbf{L}_{\mathcal{H}_t}^+$ denoting the Moore-Penrose pseudoinverse.

**Definition 2** (GraphRAG Readout Operator). The GraphRAG readout operator $\text{Readout}_\eta : \mathcal{Q} \times 2^{\mathcal{G}} \to \mathbb{R}^d$ is defined as a composition of graph neural network processing and cross-modal fusion:

$$u_t = \text{Readout}_\eta(q_t, \mathcal{H}_t) = \Psi_\eta(\text{GNN}_\eta(\mathcal{H}_t), \phi_\eta(q_t)), \tag{3}$$

where $\text{GNN}_\eta(\mathcal{H}_t) = \mathbf{W}_{\text{out}}\sigma(\mathbf{L}_{\mathcal{H}_t}\mathbf{X}_{\mathcal{H}_t}\mathbf{W}_{\text{in}} + \mathbf{b})$ with learnable parameters $\mathbf{W}_{\text{out}}, \mathbf{W}_{\text{in}}, \mathbf{b}$, node features $\mathbf{X}_{\mathcal{H}_t}$, and the fusion operator:

$$\Psi_\eta(\mathbf{g}, \mathbf{q}) = \text{softmax}\left(\frac{\mathbf{Q}_{\text{cross}}\mathbf{K}_{\text{cross}}^\top}{\sqrt{d}}\right)\mathbf{V}_{\text{cross}}, \tag{4}$$

where $\mathbf{Q}_{\text{cross}} = \mathbf{q}^\top\mathbf{W}_Q$, $\mathbf{K}_{\text{cross}} = \mathbf{g}^\top\mathbf{W}_K$, $\mathbf{V}_{\text{cross}} = [\mathbf{g}; \mathbf{q}]^\top\mathbf{W}_V$.

**Definition 3** (KG Update Operator). Given document set $D_t$ and extracted edge candidates $\widehat{E}_t = f_\theta(D_t)$ with confidence scores $\mathbf{c}_t \in [0, 1]^{|\widehat{E}_t|}$, the KG update operator $\mathcal{U}_\varsigma : \mathcal{G} \times 2^E \to \mathcal{G}$ is defined as the solution to the constrained optimization problem:

$$\mathcal{U}_\varsigma(\mathcal{G}_t, \widehat{E}_t) = \arg\max_{\mathcal{G} \in \mathcal{F}}\left[\mathcal{S}(\mathcal{G}; \mathcal{G}_t, \widehat{E}_t) + \mathcal{C}(\mathcal{G}; \widehat{E}_t) - \lambda_{\text{contr}}\mathcal{R}_{\text{contr}}(\mathcal{G})\right], \tag{5}$$

where $\mathcal{F}$ is the feasible graph space, and the objective components are:

$$\mathcal{S}(\mathcal{G}; \mathcal{G}_t, \widehat{E}_t) = \exp\left(-\kappa_s \cdot d_G(\mathcal{G}, \mathcal{G}_t)\right) \cdot \prod_{e \in \widehat{E}_t \cap E} c_e, \tag{6}$$

$$\mathcal{C}(\mathcal{G}; \widehat{E}_t) = \sum_{e \in \widehat{E}_t} c_e \cdot \mathbf{1}_{e \in E} - \tau \sum_{e \in E \setminus \widehat{E}_t} \exp(-\xi \cdot \mathrm{age}(e)), \tag{7}$$

$$\mathcal{R}_{\mathrm{contr}}(\mathcal{G}) = \sum_j \max(0, g_j(\mathcal{G}))^2, \tag{8}$$

with graph distance $d_G(\cdot, \cdot)$, edge age function $\mathrm{age}(\cdot)$, and constraint functions $g_j(\cdot)$.

**Definition 4** (Environmental Reward Components). The environmental reward incorporates two graph-theoretic measures. The coverage gain is defined via the submodular function:

$$\Delta\mathsf{Cov}_\kappa(\mathcal{G}_t \to \mathcal{G}_{t+1}) = \sum_{v \in V_{t+1}} \left[1 - (1-\kappa)^{|\mathcal{N}_{\mathcal{G}_{t+1}}(v;h)|}\right] - \sum_{v \in V_t} \left[1 - (1-\kappa)^{|\mathcal{N}_{\mathcal{G}_t}(v;h)|}\right], \tag{9}$$

where $\mathcal{N}_{\mathcal{G}}(v; h)$ denotes the $h$-hop neighborhood of vertex $v$. The von Neumann entropy gain is:

$$\Delta\mathsf{Ent}_{\mathrm{VN}}(\mathcal{G}_t \to \mathcal{G}_{t+1}) = -\mathrm{tr}\left(\rho_{t+1} \log \rho_{t+1}\right) + \mathrm{tr}\left(\rho_t \log \rho_t\right), \tag{10}$$

where $\rho_t = \frac{\mathbf{L}_{\mathcal{G}_t} + \mu\mathbf{I}}{\mathrm{tr}(\mathbf{L}_{\mathcal{G}_t} + \mu\mathbf{I})}$ is the normalized density operator. The complete environmental reward is:

$$R_{\mathrm{env}}(s_t, a_t) = \Delta\mathsf{Cov}_\kappa(\mathcal{G}_t \to \mathcal{G}_{t+1}) + \Delta\mathsf{Ent}_{\mathrm{VN}}(\mathcal{G}_t \to \mathcal{G}_{t+1}) - \lambda_{\mathrm{contr}}\mathcal{R}_{\mathrm{contr}}(\mathcal{G}_{t+1}) - \lambda_T\mathcal{T}(\mathcal{G}_t, \mathcal{G}_{t+1}). \tag{11}$$

**Definition 5** (Learnable Multi-Scale Compression). Given uncompressed embedding matrix $\mathbf{H} \in \mathbb{R}^{n \times d}$, we define $L$ learnable compression scales with adaptive attention mechanisms. Each scale $i$ computes:

$$\phi_i(\mathbf{H}; k_i) = \mathrm{softmax}\left(\frac{\mathbf{Q}_i\mathbf{K}_i^\top}{\sqrt{d/L}} + \mathbf{M}_i\right)\mathbf{V}_i \in \mathbb{R}^{k_i \times d}, \tag{12}$$

where $\mathbf{Q}_i \in \mathbb{R}^{k_i \times d}$ are learnable compression queries, $\mathbf{K}_i = \mathbf{H}\mathbf{W}_{K,i} \in \mathbb{R}^{n \times d}$, $\mathbf{V}_i = \mathbf{H}\mathbf{W}_{V,i} \in \mathbb{R}^{n \times d}$, and $\mathbf{M}_i \in \mathbb{R}^{k_i \times n}$ is a learnable bias matrix. The multi-scale compressed representation is:

$$\mathbf{Z} = \sum_{i=1}^{L} \omega_i\phi_i(\mathbf{H}; k_i) + \sum_{i=1}^{L-1}\sum_{j=i+1}^{L} \xi_{ij} \cdot \mathrm{CrossScale}(\phi_i(\mathbf{H}; k_i), \phi_j(\mathbf{H}; k_j)), \tag{13}$$

where $\omega_i \geq 0$, $\sum_i \omega_i = 1$, $\xi_{ij} \geq 0$ are cross-scale interaction weights, and:

$$\mathrm{CrossScale}(\mathbf{Z}_i, \mathbf{Z}_j) = \tanh\left(\mathbf{W}_{\mathrm{cross}}[\mathbf{P}_i\mathbf{Z}_i \odot \mathbf{P}_j\mathbf{Z}_j; \mathbf{P}_i\mathbf{Z}_i; \mathbf{P}_j\mathbf{Z}_j] + \mathbf{b}_{\mathrm{cross}}\right), \tag{14}$$

with $\odot$ denoting element-wise product, $\oplus$ concatenation after dimension alignment, and $\mathbf{W}_{\mathrm{cross}}, \mathbf{b}_{\mathrm{cross}}$ as learnable parameters.

## 2.2 Algorithm and Theory

**Agent-Knowledge Graph Co-Evolution Operator.** The co-evolution between the agent and knowledge graph is formalized through the joint operator $\Phi : (\theta_t, \mathcal{G}_t) \mapsto (\theta_{t+1}, \mathcal{G}_{t+1})$. The agent parameter update follows the policy gradient with GraphRAG-conditioned advantage estimation:

$$\theta_{t+1} = \theta_t + \eta_\theta\widehat{\nabla}_\theta J(\theta_t; \mathcal{G}_t, \mathcal{M}_t), \tag{15}$$

where the objective function $J(\theta) = \mathbb{E}_{\pi_\theta}[\sum_{t \geq 0} \gamma^t R_t]$ incorporates the dual reward mechanism. The advantage estimator $\widehat{A}_t$ utilizes the compressed observation $o_t \equiv (\mathbf{Z}_t, q_t, u_t, \mathrm{Enc}(\mathcal{G}_t))$ fused with GraphRAG readout. The knowledge graph evolves simultaneously through $\mathcal{G}_{t+1} = \mathcal{U}_\zeta(\mathcal{G}_t, f_\theta(D_t))$, creating a feedback loop where improved extraction capabilities lead to richer graph structures, which in turn enable more effective retrieval for future decisions.

Table 1: Performance evaluation on graph extraction benchmarks across different model architectures and training methodologies. The best results are highlighted in **bold**, and the second-best results are underlined.

| Model → on DATASET via METHOD | IEPile | | MmlKG | | ConfigKG | | WirelessKG | | DcommKG | |
|---|---|---|---|---|---|---|---|---|---|---|
| | NER | RE | NER | RE | NER | RE | NER | RE | NER | RE |
| Qwen2.5 7B | 59.35 | 34.28 | 44.72 | 0.10 | 98.23 | 24.17 | 64.99 | 17.42 | 27.16 | 14.16 |
| Qwen2.5 14B | 67.13 | 43.32 | 48.95 | 2.36 | 98.23 | 29.17 | 61.19 | 35.42 | 47.65 | 13.95 |
| Qwen2.5 32B | 68.05 | 46.69 | 53.90 | 3.95 | 98.23 | 60.28 | 63.59 | 45.18 | 35.78 | 17.10 |
| QwQ | 70.76 | 57.39 | 56.89 | 20.62 | 98.23 | 69.72 | 65.14 | 64.29 | 41.18 | 15.18 |
| Qwen2.5 7B → on AUTOKG via SFT | 56.89 | 36.39 | 49.50 | 0.00 | 98.23 | 29.17 | 61.93 | 12.84 | 22.42 | 9.23 |
| Qwen2.5 14B → on AUTOKG via SFT | 66.82 | 43.69 | 42.72 | 1.38 | 98.23 | 32.50 | 53.46 | 45.42 | 48.57 | 8.75 |
| Qwen2.5 32B → on AUTOKG via SFT | 68.17 | 47.87 | 54.12 | 4.39 | 98.23 | 56.95 | 56.70 | 44.02 | 33.05 | 15.62 |
| QwQ → on AUTOKG via SFT | 70.88 | 58.84 | 57.12 | 2.68 | 98.23 | 58.08 | 62.64 | 62.38 | 38.04 | 13.87 |
| Qwen2.5 7B → on COMMTKG via SFT | 46.49 | 34.47 | 51.17 | 0.00 | 98.23 | 23.54 | 69.40 | 30.67 | 31.74 | 14.28 |
| Qwen2.5 14B → on COMMTKG via SFT | 60.80 | 46.58 | 49.48 | 0.00 | 98.23 | 24.72 | 52.50 | 46.58 | 49.84 | 20.35 |
| Qwen2.5 32B → on COMMTKG via SFT | 66.46 | 44.91 | 55.32 | 8.34 | 98.23 | 39.17 | 64.46 | 35.95 | 39.66 | 4.63 |
| QwQ → on COMMTKG via SFT | 69.11 | 55.20 | 58.39 | 12.73 | 98.23 | 51.83 | 66.03 | 51.16 | 45.65 | 4.11 |
| Qwen2.5 7B → on AUTOKG via RL | 60.99 | 45.28 | 50.55 | 0.00 | 98.23 | 32.50 | 47.58 | 8.63 | 28.22 | 5.00 |
| Qwen2.5 14B → on AUTOKG via RL | 69.32 | 53.02 | 53.37 | 0.13 | 98.23 | 32.50 | 54.10 | 39.08 | 56.40 | 13.87 |
| Qwen2.5 32B → on AUTOKG via RL | 70.15 | 54.83 | 56.77 | 1.61 | 98.23 | 39.17 | 71.84 | 49.62 | 37.20 | 9.09 |
| QwQ → on AUTOKG via RL | 69.04 | 67.40 | 59.92 | 8.37 | 98.23 | 63.03 | 73.59 | **70.61** | 42.81 | 8.07 |
| Qwen2.5 7B → on COMMTKG via RL | 67.77 | 42.87 | 46.43 | 0.00 | 98.23 | 19.17 | 65.93 | 38.07 | 32.57 | 18.70 |
| Qwen2.5 14B → on COMMTKG via RL | 70.57 | 58.35 | 59.26 | 0.77 | 98.23 | 32.50 | 60.20 | 46.39 | 61.15 | **22.94** |
| Qwen2.5 32B → on COMMTKG via RL | 71.41 | 56.79 | 63.11 | 9.32 | 98.23 | 72.50 | 67.56 | 53.06 | 45.97 | 20.16 |
| QwQ → on COMMTKG via RL | 71.66 | 69.80 | 66.61 | 37.78 | 98.23 | 69.21 | 75.50 | 52.91 | 53.48 | 17.90 |
| Qwen2.5 7B → on AUTOKG via AGENTIC-KGR (ours) | 65.11 | 45.25 | 54.90 | 0.10 | 98.23 | 24.17 | 65.71 | 16.79 | 27.14 | 6.22 |
| Qwen2.5 14B → on AUTOKG via AGENTIC-KGR (ours) | 72.76 | 55.94 | 52.84 | 0.60 | 98.23 | 34.17 | 57.25 | 44.05 | 49.80 | 16.45 |
| Qwen2.5 32B → on AUTOKG via AGENTIC-KGR (ours) | **75.08** | 54.50 | 57.15 | 3.38 | 98.23 | 72.50 | 54.34 | 39.76 | 24.61 | 15.26 |
| QwQ → on AUTOKG via AGENTIC-KGR (ours) | 71.99 | 68.73 | 64.32 | 18.23 | 98.23 | 65.66 | 76.58 | 68.32 | 40.27 | 13.55 |
| Qwen2.5 7B → on COMMTKG via AGENTIC-KGR (ours) | 64.13 | 45.76 | 50.99 | 0.00 | 98.23 | 29.17 | 67.53 | 37.85 | 29.69 | 19.94 |
| Qwen2.5 14B → on COMMTKG via AGENTIC-KGR (ours) | 66.72 | 56.69 | 58.92 | 0.64 | 98.23 | 35.83 | 62.05 | 48.63 | **62.08** | 22.45 |
| Qwen2.5 32B → on COMMTKG via AGENTIC-KGR (ours) | 70.77 | 59.09 | 64.71 | 11.82 | 98.23 | 70.28 | 71.84 | 53.92 | 42.29 | 22.56 |
| QwQ → on COMMTKG via AGENTIC-KGR (ours) | 73.59 | **72.63** | **68.30** | **46.63** | 98.23 | **73.59** | 76.73 | 48.67 | 55.83 | 20.03 |

**Environmental Reward and Adaptive Reward Mixing.** The environmental component $R_{\text{env}}$ promotes healthy graph growth by rewarding coverage expansion and structural diversity while penalizing constraint violations. We incorporate a temporal consistency regularizer $\mathcal{T}(\mathcal{G}_t, \mathcal{G}_{t+1}) = \exp(-\beta_t \cdot \|\mathbf{L}_{\mathcal{G}_{t+1}} - \mathbf{L}_{\mathcal{G}_t}\|_F)$ that prevents abrupt structural changes.

In Agentic-KGR training, the reward structure comprises three components: result reward, toolcall reward, and trajectory reward. The toolcall reward incentivizes successful tool interactions with $+0.05$ for successful calls and $-0.1$ for failures, applying decay for redundant calls within the same query and capping at $0.5$. The result reward combines format compliance ($\pm 1.0$ for JSON structure adherence), accuracy measurement (full match yields $+1.5$, partial match measured by F1 score), and density penalty that penalizes length deviations as $\||A| - |B|\|/|B| \times$ rate where rate is $0.15$ for over-generation and $0.8$ for under-generation. Trajectory reward captures multi-step interaction quality.

The mixing parameter $\alpha$ adapts dynamically via mirror descent on the bi-criterion Pareto frontier:

$$\alpha_{t+1} = \Pi_{[0,1]}\left(\alpha_t + \eta_\alpha \left[\nabla_\alpha J_{\text{env}}(\alpha_t) - \nabla_\alpha J_{\text{task}}(\alpha_t)\right]^\top \mathbf{1}\right), \tag{16}$$

where the gradients are estimated using policy gradient techniques, enabling automatic balance between environmental exploration and task-specific exploitation as the agent learns to optimize both tool usage efficiency and extraction quality.

**Multi-Scale Prompt Compression.** The compression objective jointly optimizes reconstruction fidelity, task performance, and information retention:

$$\mathcal{L}_{\text{compress}} = \lambda_{\text{rec}}\|\mathbf{H} - \psi(\mathbf{Z})\|_F^2 + \lambda_{\text{task}}\mathcal{L}_{\text{KG}}(\mathbf{Z}) + \lambda_{\text{MI}}I(\mathbf{H}; \mathbf{Z}), \tag{17}$$

where $\psi : \mathbb{R}^{k_{\text{eff}} \times d} \to \mathbb{R}^{n \times d}$ is a learned decompression function implemented as a transformer decoder with cross-attention to the compressed representation. The mutual information term $I(\mathbf{H}; \mathbf{Z})$ is estimated using the Donsker-

Table 2: Performance comparison of different model architectures and training strategies with GraphRAG across seven real-world QA tasks. The best results are highlighted in **bold**, and the second-best results are underlined.

| Model → on DATASET via METHOD + GraphRAG | RAN FDD | BWS | MA5600T | OptiTran | LineAss | NetEco | PowerKit |
|---|---|---|---|---|---|---|---|
| Qwen2.5 7B + GraphRAG | 61.54 | 50.60 | 52.30 | 57.90 | 48.64 | 67.12 | 64.94 |
| Qwen2.5 14B + GraphRAG | 78.08 | 71.92 | 78.46 | 75.26 | 74.32 | 70.06 | 77.28 |
| Qwen2.5 32B + GraphRAG | 79.24 | 72.28 | 75.38 | 84.22 | 70.28 | 71.14 | 83.12 |
| QwQ + GraphRAG | 61.54 | 43.38 | 60.00 | 59.64 | 69.16 | 61.74 | 70.06 |
| Qwen2.5 7B → on AUTOKG via SFT + GraphRAG | 53.84 | 54.60 | 55.38 | 59.70 | 46.84 | 70.72 | 68.54 |
| Qwen2.5 14B → on AUTOKG via SFT + GraphRAG | 70.00 | 67.10 | 85.38 | 82.28 | 79.72 | 86.04 | 79.48 |
| Qwen2.5 32B → on AUTOKG via SFT + GraphRAG | 89.24 | 79.46 | 83.84 | 85.96 | **90.28** | 87.12 | 90.52 |
| QwQ → on AUTOKG via SFT + GraphRAG | 57.80 | 51.34 | 59.56 | 57.84 | 72.52 | 70.42 | 69.56 |
| Qwen2.5 7B → on COMMTKG via SFT + GraphRAG | 76.54 | 62.66 | 83.08 | 72.64 | 77.84 | 88.32 | 81.56 |
| Qwen2.5 14B → on COMMTKG via SFT + GraphRAG | 84.62 | 69.52 | 91.30 | 81.22 | 87.33 | 85.90 | 88.79 |
| Qwen2.5 32B → on COMMTKG via SFT + GraphRAG | 85.48 | 75.78 | 96.16 | **89.12** | 87.84 | 87.58 | 87.56 |
| QwQ → on COMMTKG via SFT + GraphRAG | 73.46 | 70.96 | 78.46 | 76.66 | 77.30 | 78.46 | 77.92 |
| Qwen2.5 7B → on AUTOKG via RL + GraphRAG | 59.62 | 57.84 | 63.08 | 63.16 | 65.68 | 69.80 | 72.72 |
| Qwen2.5 14B → on AUTOKG via RL + GraphRAG | 78.08 | 77.10 | 87.76 | 76.50 | 83.52 | 84.70 | 81.68 |
| Qwen2.5 32B → on AUTOKG via RL + GraphRAG | 81.54 | 73.98 | 80.76 | 84.04 | 78.10 | 84.10 | 84.68 |
| QwQ → on AUTOKG via RL + GraphRAG | 55.96 | 63.64 | 58.26 | 69.92 | 76.84 | 84.36 | 77.26 |
| Qwen2.5 7B → on COMMTKG via RL + GraphRAG | 82.30 | 69.88 | 80.76 | 71.92 | 81.52 | 89.96 | 78.70 |
| Qwen2.5 14B → on COMMTKG via RL + GraphRAG | 83.84 | 69.88 | 86.92 | 82.40 | 88.92 | 87.82 | 89.48 |
| Qwen2.5 32B → on COMMTKG via RL + GraphRAG | 86.54 | 77.10 | 97.70 | 84.51 | 87.32 | 85.30 | 91.90 |
| QwQ → on COMMTKG via RL + GraphRAG | 85.38 | 75.42 | 78.46 | 68.42 | 78.30 | 79.80 | 83.12 |
| Qwen2.5 7B → on AUTOKG via AGENTIC-KGR + GraphRAG (ours) | 60.32 | 63.02 | 61.54 | 64.23 | 64.06 | 70.08 | 74.94 |
| Qwen2.5 14B → on AUTOKG via AGENTIC-KGR + GraphRAG (ours) | 82.76 | 76.16 | 83.27 | 83.50 | 88.10 | 82.40 | 82.14 |
| Qwen2.5 32B → on AUTOKG via AGENTIC-KGR + GraphRAG (ours) | 86.16 | **82.28** | 82.31 | 82.72 | 79.46 | 76.38 | 87.54 |
| QwQ → on AUTOKG via AGENTIC-KGR + GraphRAG (ours) | 57.70 | 66.73 | 62.30 | 68.42 | 79.32 | 87.72 | 77.92 |
| Qwen2.5 7B → on COMMTKG via AGENTIC-KGR + GraphRAG (ours) | 83.84 | 70.65 | 81.54 | 73.14 | 84.69 | **90.40** | 80.12 |
| Qwen2.5 14B → on COMMTKG via AGENTIC-KGR + GraphRAG (ours) | 83.20 | 77.52 | 89.24 | 85.26 | 88.14 | 86.04 | 89.10 |
| Qwen2.5 32B → on COMMTKG via AGENTIC-KGR + GraphRAG (ours) | **91.54** | 77.46 | **98.46** | 84.22 | **90.28** | 87.12 | **92.72** |
| QwQ → on COMMTKG via AGENTIC-KGR + GraphRAG (ours) | 87.35 | 72.89 | 81.23 | 70.12 | 79.80 | 81.30 | 82.14 |

Varadhan variational representation with a discriminator network. Additionally, we impose a trust-region style constraint $\mathbb{E}_s[\mathrm{KL}(\pi_\theta(\cdot \mid \mathbf{Z}(s))\|\pi_\theta(\cdot \mid \mathbf{H}(s)))] \leq \epsilon_\pi$ to ensure policy stability under compression, preventing catastrophic forgetting of learned behaviors while enabling efficient inference.

To ensure that prompt compression does not significantly deteriorate agent performance, we establish theoretical guarantees on the performance gap between compressed and uncompressed policies. Let $\pi_H(a \mid s) \equiv \pi_\theta(a \mid \mathbf{H}(s))$ and $\pi_Z(a \mid s) \equiv \pi_\theta(a \mid \mathbf{Z}(s))$ denote policies operating on uncompressed and compressed observations respectively. Under Lipschitz continuity assumptions on the dynamics, rewards, and policy function, the performance degradation is bounded by:

$$|J(\pi_Z) - J(\pi_H)| \leq \frac{L_R}{1-\gamma}\varepsilon_{\mathrm{obs}} + \frac{\gamma L_Q \sigma_\pi}{(1-\gamma)^2}\varepsilon_{\mathrm{obs}} + \frac{2Q_{\max}}{1-\gamma}\sqrt{\frac{1}{2}\epsilon_\pi}, \tag{18}$$

where $\varepsilon_{\mathrm{obs}} = \mathbb{E}_s[\|\mathbf{H}(s) - \psi(\mathbf{Z}(s))\|_2]$ measures the observation reconstruction error, $\epsilon_\pi = \mathbb{E}_s[\mathrm{KL}(\pi_\theta(\cdot \mid \mathbf{Z}(s))\|\pi_\theta(\cdot \mid \mathbf{H}(s)))]$ quantifies the policy distribution shift, and $L_R, L_Q, \sigma_\pi, Q_{\max}$ are problem-dependent constants. This bound demonstrates that the performance loss scales linearly with reconstruction error and as $\sqrt{\epsilon_\pi}$ with policy divergence, justifying our compression constraints and providing guidance for hyperparameter selection.

# 3 EXPERIMENTS AND RESULTS

## 3.1 ENVIRONMENT AND CONFIGURATION

**Platform and Environment.** All experiments are conducted on 128 Ascend 910B3 NPUs (64GB each) with driver version 25.0.rc1. We employ three training frameworks: MindSpeed-LLM (min, 2023) for supervised fine-tuning, MindSpeed-RL (Feng et al., 2025) for standard single-turn RL, and our custom MindSpeed-AgenticRL framework (open-source release in preparation) for Agentic-KGR. All training processes utilize bf16 precision for computational efficiency and numerical stability.

**Training Parameters.** The key hyperparameters for each training paradigm are configured as follows: SFT employs learning rate $1 \times 10^{-6}$, global batch size 32, and sequence length 8k; standard RL uses learning rate $1 \times 10^{-6}$, 8 samples per prompt, and GAE lambda 0.95; Agentic RL maintains maximum interaction steps 10, 8 samples per prompt, and extends sequence length to 16k to accommodate multi-round agent-environment interactions.

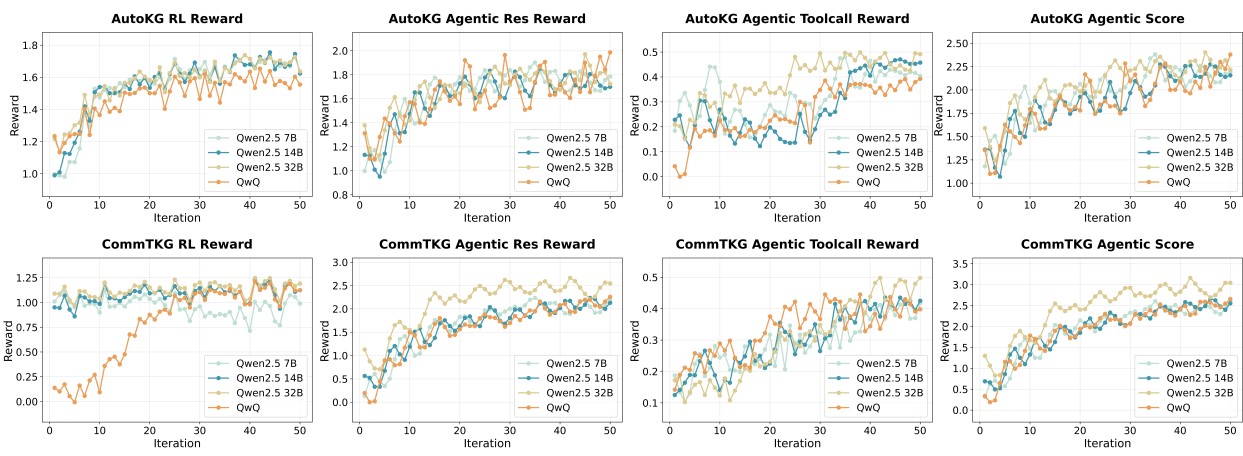

Figure 3: Training reward variation for RL and Agentic-KGR methods across training steps.

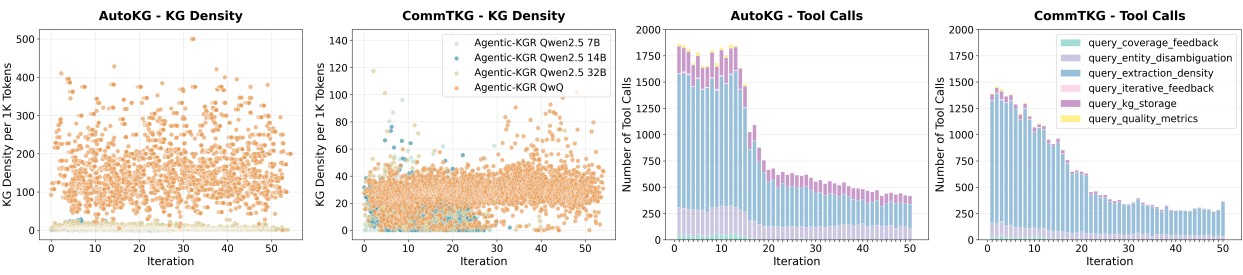

Figure 5: Graph density analysis and tool invocation frequency distribution.

**Training Datasets.** Training data comprises general-domain and domain-specific datasets. For general knowledge extraction, we utilize DuIE2.0 from the AutoKG corpus (Chen & Bertozzi, 2023), while domain-specific training leverages CommTKG encompassing MML (Man–Machine Language, an ITU-T–standardized command language for managing telecom/network equipment), communcatioan technology product/feature documentation. The corpus integrates authoritative standards from IEEE (iee, 2024), OMA SpecWorks (oma, 2024), 3GPP (3gp, 2024), nrexplained (nre, 2024), and CTIA (cti, 2024). Ground truth annotations are generated through DeepSeek V3.1 distillation following systematic document parsing and segmentation.

**Benchmark and Metrics.** Evaluation encompasses dual assessment dimensions to validate the co-evolutionary effectiveness: KG extraction capability and GraphRAG-based knowledge coverage quality. For extraction evaluation, general benchmarks include IEPile (Gui et al., 2024) covering Named-entity recognition (NER) tasks (Boson, Cross, WEIBONER) and Relation extraction (RE) tasks (COAE2016, FewRel), while domain-specific evaluation employs MmlKG, ConfigKG, WirelessKG, and DcommKG across corresponding NER and RE tasks. For QA tasks, we adopt the QA pairs from communication technology product documentations, containing: 5G RAN FDD (RAN FDD), BWS, MA5600T, Optical Transmission System (Opti-Tran), Line Assurance Platform (LineAss), NetECo and PowerKit.

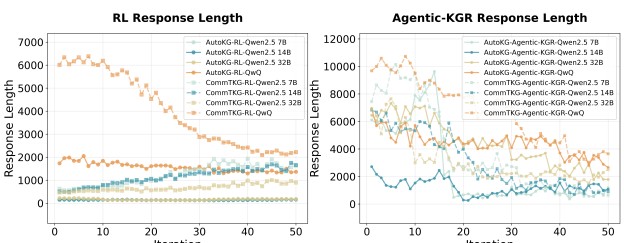

Figure 4: Response length evolution during RL and Agentic-KGR training across different model scales.

To align with operational requirements, all evaluations adopt F1 score as the primary metric, emphasizing comprehensive knowledge capture over precision constraints.

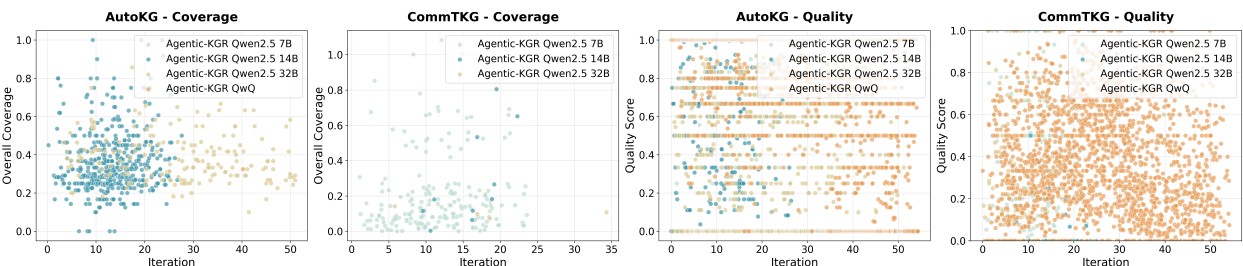

Figure 6: Coverage and quality performance analysis.

## 3.2 OVERALL RESULTS

### 3.2.1 GRAPH EXTRACTION PERFORMANCE

We evaluate graph extraction capabilities across different model configurations and training methodologies, as shown in Table 1. The results reveal systematic improvements with model scale and, more critically, demonstrate that Agentic-KGR achieves superior performance through co-evolutionary parameter-knowledge optimization. The NER task on ConfigKG proves relatively straightforward, with all models achieving consistently high performance scores. Notably, gains are most pronounced in RE rather than NER, reflecting our method's design: dynamic schema expansion increases relational discoverability while retrieval-augmented memory reduces spurious connections. The learnable multi-scale prompt compression mechanism enables models to focus on schema-salient evidence patterns, contributing to improved extraction quality across benchmark datasets. Performance plateaus observed in single-round RL are consistently overcome by multi-round, memory-coupled policy updates, validating the co-evolution hypothesis across model scales from 7B to 32B parameters.

### 3.2.2 END-TO-END QA PERFORMANCE

In Table 2, the downstream QA evaluation confirms that graph quality, not merely size, drives performance when integrated with GraphRAG. Agentic-KGR configurations achieve dominant performance across most domains, with gains being largest where schema breadth and cross-document linking are crucial. The improvements are not merely additive with parameter count; rather, Agentic updates interact synergistically with GraphRAG to reduce retrieval misses and mitigate hallucination through denser, better-typed knowledge neighborhoods. These quality improvements manifest as higher QA accuracy through more effective retrieval chains and answer grounding. The results validate that structured, relation-centric improvements in extraction translate directly into downstream task gains when the complete workflow is optimized for parameter-knowledge co-evolution.

## 3.3 TRAINING DYNAMICS AND EFFICIENCY ANALYSIS

The training curves reveal critical insights into model-task alignment and co-evolutionary learning dynamics, as shown in Figure 3. QwQ's initially low rewards demonstrate that reasoning-intensive architectures can impede structured extraction tasks, where excessive deliberation interferes with direct entity-relation pattern recognition. The steady progression across all reward components validates the effectiveness of our co-evolution mechanism: toolcall rewards show consistent improvement, indicating successful adaptation in retrieval strategy selection that creates a positive feedback loop with content generation. The alignment between agentic score and response rewards, coupled with smooth convergence across model scales, confirms that our multi-scale prompt compression successfully manages optimization complexity while preserving critical information pathways essential for dynamic schema expansion.

The response length trajectories reveal distinct optimization patterns across training paradigms (see Figure 4). Standard RL effectively curtails excessive reasoning in thinking models, with QwQ showing dramatic reduction from 6k to 2k tokens on domain datasets, while non-thinking models exhibit slight increases indicating enhanced extraction density. Agentic-KGR demonstrates superior optimization, achieving consistent length reduction across all architectures through progressive interaction efficiency gains. The universal downward trends under Agentic-KGR validate that co-evolutionary optimization successfully streamlines extraction processes while suppressing verbose reasoning patterns, confirming the synergistic effects of retrieval-augmented memory and adaptive schema expansion in achieving both efficiency and structural completeness.

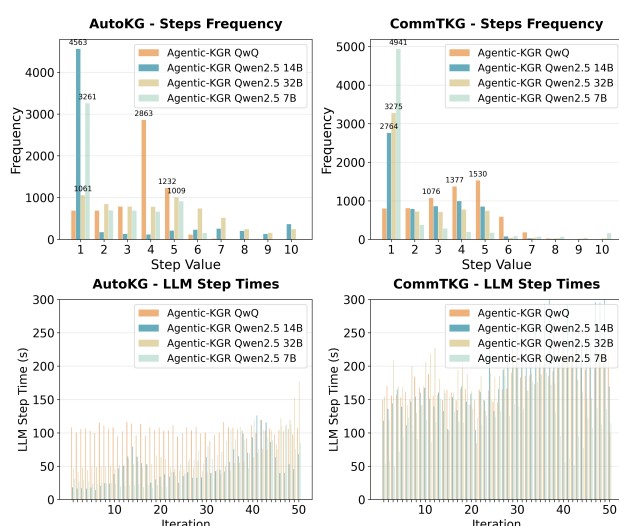

Figure 7: Response length distribution analysis across model variants and training methods.

## 3.4 KNOWLEDGE STRUCTURE ANALYSIS

The density, coverage, and quality metrics displayed in Figure 5 and Figure 6 demonstrate distinct optimization trajectories across model architectures. Thinking models exhibit higher tool interaction frequencies, enabling comprehensive data collection with progressive improvement throughout training, particularly on domain-specific datasets. The agentic framework successfully balances exploration (coverage) and precision (quality), with larger models achieving superior performance and quality scores approaching optimal levels on specialized domains. Tool usage analysis reveals density feedback queries dominate initial interactions but decline across iterations, indicating improved extraction efficiency through adaptive learning. This trend reflects how models initially require frequent guidance but progressively achieve target thresholds with fewer tool interventions, validating the retrieval-augmented memory system's effectiveness in internalizing structural optimization patterns while reducing computational overhead.

Thinking models exhibit inherently verbose reasoning patterns, but Agentic-KGR effectively compresses these outputs while preserving extraction quality through

Figure 8: Step distribution analysis and computational overhead across model variants.

multi-scale prompt optimization (see Figure 7). The framework demonstrates superior efficiency gains on domain-specific tasks, validating the co-evolutionary approach's ability to balance reasoning depth with output conciseness. In Figure 8, the step distribution analysis reveals that smaller models favor direct problem-solving approaches while larger models employ deeper deliberative reasoning with extended multi-step processes. The computational overhead remains stable across iterations, demonstrating the framework's ability to dynamically balance reasoning complexity with computational efficiency based on task requirements.

## 4 CONCLUSION

This work proposes *Agentic-KGR*, a novel framework enabling co-evolution between language models and dynamic knowledge graphs through multi-round reinforcement learning. The dual reward mechanism achieves significant performance improvements by enabling autonomous learning of effective graph database interaction patterns, while the retrieval-augmented memory mechanism continuously optimizes knowledge graph construction and provides comprehensive graph observation during training. The framework transcends traditional static knowledge base limitations by empowering models with real-time knowledge extraction, construction, and expansion capabilities, with integrated GraphRAG retrieval demonstrating superior performance in downstream question-answering tasks. This research establishes a new paradigm for model-environment co-evolution that advances agentic reinforcement learning by demonstrating how continuous interaction between intelligent agents and dynamic knowledge environments can mutually enhance both components.

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

# A APPENDIX

## A.1 ACKNOWLEDGEMENTS

The authors acknowledge the use of Claude Sonnet 4 for stylistic revision, text polishing, and grammatical refinement of this manuscript, with full consent and approval from all contributing authors. We emphasize that the substantive content, including conceptual design, theoretical framework development, experimental methodology, algorithmic implementation, and code development, was entirely conceived and executed by the authors. The AI assistance was limited to language enhancement and does not constitute intellectual contribution to the research findings or technical innovations presented in this work.

## A.2 RELATED WORK

### A.2.1 RETRIEVAL-AUGMENTED GENERATION AND KNOWLEDGE ENHANCEMENT

RAG has emerged as a fundamental paradigm for addressing static knowledge limitations in LLMs. Gao et al. (2024) categorize approaches into Naive RAG, Advanced RAG, and Modular RAG, while identifying persistent challenges in knowledge coverage and temporal relevance. Traditional RAG systems rely on pre-constructed knowledge bases, which suffer from coverage gaps and temporal lag issues.

Recent advances have integrated web search capabilities to overcome static limitations. Li et al. (2025a) introduce Search-o1 with agentic RAG mechanisms, while Li et al. (2025b) propose WebThinker for autonomous web navigation and report generation. However, these approaches primarily focus on information retrieval without addressing knowledge persistence or structural evolution of underlying knowledge bases.

Evaluation frameworks including GAIA (Mialon et al., 2024) and WebWalkerQA (Wu et al., 2025b) reveal performance gaps in complex reasoning tasks, highlighting the need for more sophisticated knowledge integration approaches.

### A.2.2 REINFORCEMENT LEARNING FOR KNOWLEDGE-INTENSIVE TASKS

RL applications to enhance LLM capabilities in knowledge-intensive scenarios have gained momentum. Jin et al. (2025) introduce Search-R1 for autonomous search query generation using outcome-based rewards, while Song et al. (2025a) and Song et al. (2025b) develop R1-Searcher and R1-Searcher++ focusing on external search system integration.

Several works explore step-wise supervision to address sparse reward challenges. Wang et al. (2025) propose StepSearch with step-wise proximal policy optimization, Deng et al. (2025) introduce Atom-Searcher with fine-grained atomic rewards, and Chen et al. (2025) present ReSearch treating search operations as reasoning components. These approaches demonstrate the importance of dense reward signals.

Current RL-based approaches primarily optimize search and retrieval behaviors rather than enabling dynamic KG construction. The knowledge bases remain static, limiting their ability to capture emerging information or domain-specific knowledge arising during interaction.

### A.2.3 MULTI-AGENT SYSTEMS AND COLLABORATIVE LEARNING

Multi-agent frameworks have shown promise in complex task solving through specialized coordination. Zhang et al. (2025) introduce AgentOrchestra with hierarchical conductor-orchestra dynamics, while Wu et al. (2025a) present WebDancer for autonomous information seeking through multi-step reasoning.

Recent research explores multi-agent RL for LLM collaboration. Liao et al. (2025) propose MARFT with POMDP formulations, Liu et al. (2025) develop MAGRPO for cooperative scenarios, Wan et al. (2025) introduce ReMA for meta-thinking hierarchies, and Gao et al. (2025) propose FlowReasoner for query-level meta-agent design.

These multi-agent approaches demonstrate effective collaboration patterns but operate within fixed knowledge environments where underlying knowledge structures do not evolve during interaction, constraining adaptability to emerging information needs. Current approaches either focus on static knowledge retrieval, dynamic search without knowledge persistence, or multi-agent collaboration within fixed knowledge bounds. Few works address co-evolution between reasoning models and their underlying knowledge structures, leaving a gap in enabling adaptive knowledge-model systems that continuously evolve through interaction.

### A.3 SUPPLEMENTARY THEORY AND MISSING PROOF

**Theorem A.1** (Performance Degradation Bound under Compression). *Let $\pi_H(a \mid s) \equiv \pi_\theta(a \mid \mathbf{H}(s))$ and $\pi_Z(a \mid s) \equiv \pi_\theta(a \mid \mathbf{Z}(s))$ with $\mathbf{Z} = \phi(\mathbf{H})$. Under $\gamma \in (0, 1)$, the performance gap satisfies:*

$$\left| J(\pi_Z) - J(\pi_H) \right| \leq \frac{L_R}{1 - \gamma} \varepsilon_{\text{obs}} + \frac{\gamma L_Q \sigma_\pi}{(1 - \gamma)^2} \varepsilon_{\text{obs}} + \frac{2Q_{\max}}{(1 - \gamma)} \sqrt{\tfrac{1}{2} \epsilon_\pi},$$

*where $\varepsilon_{\text{obs}} \triangleq \mathbb{E}_s \|\mathbf{H}(s) - \psi(\phi(\mathbf{H}(s)))\|_2 \leq \varepsilon_{\text{rec}}$ and $\epsilon_\pi \triangleq \mathbb{E}_s \mathrm{KL}\big(\pi_\theta(\cdot \mid \mathbf{Z}(s)) \,\|\, \pi_\theta(\cdot \mid \mathbf{H}(s))\big)$.*

*Remark* A.1. The bound demonstrates that performance loss scales linearly with reconstruction error and as $\sqrt{\epsilon_\pi}$ with policy KL-divergence. This justifies our compression constraints and provides principled guidance for selecting the trade-off parameters $\lambda_{\text{rec}}$, $\lambda_{\text{MI}}$, and $\epsilon_\pi$.

*Proof of Theorem A.1.* We apply the performance difference lemma to decompose the gap between compressed and uncompressed policies. The fundamental identity states:

$$J(\pi_Z) - J(\pi_H) = \frac{1}{1 - \gamma} \mathbb{E}_{s \sim d^{\pi_Z}} \mathbb{E}_{a \sim \pi_Z} \left[ A^{\pi_H}(s, a) \right],$$

where $d^{\pi_Z}$ is the stationary distribution under $\pi_Z$ and $A^{\pi_H}(s, a) = Q^{\pi_H}(s, a) - V^{\pi_H}(s)$ is the advantage function under $\pi_H$. We bound this expression by decomposing the advantage discrepancy and state distribution shift.

First, we bound the advantage function difference. $Q^{\pi_H}(s, a)$ is $L_Q$-Lipschitz in $s$. For observations $s$ and $\tilde{s}$ differing only in their token representations ($\mathbf{H}(s)$ vs. $\mathbf{Z}(s) = \phi(\mathbf{H}(s))$), we have:

$$|A^{\pi_H}(s, a) - A^{\pi_H}(\tilde{s}, a)| \leq L_Q \|\mathbf{H}(s) - \psi(\mathbf{Z}(s))\|_2 = L_Q \varepsilon_{\text{obs}}.$$

Second, we address the policy distribution mismatch. By Pinsker's inequality and the definition of $\epsilon_\pi$:

$$\mathrm{TV}(\pi_Z(\cdot|s), \pi_H(\cdot|s)) \leq \sqrt{\tfrac{1}{2} \mathrm{KL}(\pi_Z(\cdot|s) \| \pi_H(\cdot|s))} \leq \sqrt{\tfrac{1}{2} \epsilon_\pi}.$$

Since $|A^{\pi_H}(s, a)| \leq 2Q_{\max}$ by boundedness, the contribution from action distribution mismatch is bounded by $2Q_{\max} \sqrt{\epsilon_\pi / 2}$.

Third, we bound the state distribution shift. Under compressed policy $\pi_Z$, the observation at each step has expected error $\varepsilon_{\text{obs}}$. By $\pi_\theta$ is $\sigma_\pi$-Lipschitz in observations and dynamics are $L_P$-Lipschitz, the state distribution difference satisfies:

$$\|d^{\pi_Z} - d^{\pi_H}\|_1 \leq \frac{\gamma L_P \sigma_\pi \varepsilon_{\text{obs}}}{1 - \gamma}.$$

Combining these bounds and telescoping the expectation over the shifted distribution:

$$|J(\pi_Z) - J(\pi_H)| \leq \frac{1}{1 - \gamma} \left[ L_R \varepsilon_{\text{obs}} + \gamma L_Q \sigma_\pi \varepsilon_{\text{obs}} \cdot \frac{1}{1 - \gamma} + 2Q_{\max} \sqrt{\frac{\epsilon_\pi}{2}} \right] \tag{19}$$

$$= \frac{L_R \varepsilon_{\text{obs}}}{1 - \gamma} + \frac{\gamma L_Q \sigma_\pi \varepsilon_{\text{obs}}}{(1 - \gamma)^2} + \frac{2Q_{\max}}{1 - \gamma} \sqrt{\frac{\epsilon_\pi}{2}}, \tag{20}$$

where the first term captures direct reward sensitivity to observation errors, the second accounts for compounded effects through state distribution shift, and the third bounds the policy mismatch contribution. $\square$ $\square$

**Theorem A.2** (Policy Improvement with Trust Region under Compression). *Let $\pi'$ solve the graph-regularized surrogate maximization problem:*

$$\max_\pi \quad \mathbb{E}_{s \sim d^{\pi_{\text{old}}}, a \sim \pi}[A^{\pi_{\text{old}}}(s, a)] - \lambda_{\text{graph}} \mathcal{R}_{\text{graph}}(\pi, \mathcal{G}_t)$$

$$s.t. \quad \mathbb{E}_s \mathrm{KL}(\pi(\cdot \mid s) \| \pi_{\text{old}}(\cdot \mid s)) \leq \epsilon_{\text{TR}},$$

*where $\mathcal{R}_{\text{graph}}(\pi, \mathcal{G}_t)$ is a graph-structure regularizer. If the compression constraint $\epsilon_\pi \leq \epsilon_{\text{TR}}$ holds and the surrogate is optimized to $\varepsilon$-accuracy, then:*

$$J(\pi') \geq J(\pi_{\text{old}}) + \frac{1}{1 - \gamma} \left( \mathbb{E}_{s,a}[A^{\pi_{\text{old}}}(s, a)] - \lambda_{\text{graph}} \mathcal{R}_{\text{graph}}(\pi', \mathcal{G}_t) - \mathcal{O}\big(\sqrt{\epsilon_{\text{TR}}} + \varepsilon\big) \right).$$

*Remark* A.2. This result extends classical trust region methods to the graph-regularized setting by incorporating structural constraints from the evolving knowledge graph. Unlike standard TRPO, our GRPO approach explicitly accounts for the graph topology in policy updates, ensuring that the learned policy respects the relational structure encoded in $\mathcal{G}_t$. The compression can be viewed as an additional perturbation that must be managed within the trust region framework.

*Proof of Theorem A.2.* We extend the classical trust region analysis to incorporate graph regularization and compression effects. The proof follows three main steps: establishing the surrogate objective bound, controlling the graph regularization term, and accounting for compression-induced policy drift.

First, we establish the connection between the true objective and the graph-regularized surrogate. Let $L(\pi) = \mathbb{E}_{s \sim d^{\pi_{\text{old}}}, a \sim \pi}[A^{\pi_{\text{old}}}(s,a)] - \lambda_{\text{graph}}\mathcal{R}_{\text{graph}}(\pi, \mathcal{G}_t)$ denote the surrogate objective. By the policy improvement lemma and the definition of advantage functions:

$$J(\pi') - J(\pi_{\text{old}}) = \frac{1}{1-\gamma}\mathbb{E}_{s \sim d^{\pi'}}\mathbb{E}_{a \sim \pi'}\left[A^{\pi_{\text{old}}}(s,a)\right].$$

The key insight is to decompose this expectation using importance sampling and bound the distribution mismatch terms.

Second, we bound the state distribution shift $\|d^{\pi'} - d^{\pi_{\text{old}}}\|_1$. Under the KL constraint, the total variation distance between policies is bounded by $\sqrt{2\epsilon_{\text{TR}}}$. By Lipschitz dynamics, this translates to a state distribution shift of $\mathcal{O}(\sqrt{\epsilon_{\text{TR}}})/(1-\gamma)$. The graph regularization term $\mathcal{R}_{\text{graph}}(\pi', \mathcal{G}_t)$ provides additional stability by constraining policy changes to respect graph structure.

Third, we account for compression effects. The compressed policy $\pi'$ operates on observations $\mathbf{Z}(s)$ rather than $\mathbf{H}(s)$. Since $\epsilon_\pi \leq \epsilon_{\text{TR}}$, the total KL budget is partitioned between policy updates and compression drift. The surrogate maximization ensures that within the remaining budget $\epsilon_{\text{TR}} - \epsilon_\pi$, we achieve $\varepsilon$-optimal improvement.

Combining these bounds and using the assumption that the surrogate is optimized to $\varepsilon$-accuracy:

$$J(\pi') \geq J(\pi_{\text{old}}) + \frac{1}{1-\gamma}\left[L(\pi') - \mathcal{O}\left(\sqrt{\epsilon_{\text{TR}}}\right)\right] \tag{21}$$

$$\geq J(\pi_{\text{old}}) + \frac{1}{1-\gamma}\left[\mathbb{E}_{s,a}[A^{\pi_{\text{old}}}(s,a)] - \lambda_{\text{graph}}\mathcal{R}_{\text{graph}}(\pi', \mathcal{G}_t) - \mathcal{O}(\sqrt{\epsilon_{\text{TR}}} + \varepsilon)\right], \tag{22}$$

where the graph regularization term appears explicitly in the bound, distinguishing GRPO from standard trust region methods. $\square$ $\blacksquare$

**Theorem A.3** (Submodular Coverage and Near-Optimal Growth). *Suppose the environmental reward includes coverage gain $\Delta\text{Cov}_\kappa(\mathcal{G}_t \to \mathcal{G}_{t+1})$ and actions select a bounded-size set of edges/nodes to add each round. If the policy greedily maximizes the expected marginal gain under $\mathbb{P}_\eta(\mathcal{H}_t \mid q_t)$, then per round it achieves a $(1 - 1/e)$-approximation to the optimal expected coverage increase.*

*Remark* A.3. The submodularity of $\text{Cov}_\kappa$ in selected neighborhoods enables classical approximation guarantees. This aligns RL credit assignment with principled graph expansion, ensuring that the environmental reward promotes systematic knowledge growth rather than myopic local improvements.

*Proof of Theorem A.3.* We establish the approximation guarantee by leveraging the submodular structure of the coverage function $\text{Cov}_\kappa(\mathcal{G})$ and applying the classical greedy algorithm analysis to the expected setting.

First, we verify submodularity. For any two graphs $\mathcal{G} \subseteq \mathcal{G}'$ and any additional edge set $E_{\text{new}}$, the coverage function satisfies:

$$\text{Cov}_\kappa(\mathcal{G} \cup E_{\text{new}}) - \text{Cov}_\kappa(\mathcal{G}) \geq \text{Cov}_\kappa(\mathcal{G}' \cup E_{\text{new}}) - \text{Cov}_\kappa(\mathcal{G}').$$

This follows from the definition $\text{Cov}_\kappa(\mathcal{G}) = \sum_{v \in V}[1 - (1-\kappa)^{|\mathcal{N}_\mathcal{G}(v;h)|}]$, where the marginal contribution of each vertex decreases as its neighborhood size increases, due to the concave nature of $1 - (1-\kappa)^x$ for $\kappa \in (0,1)$.

Second, we analyze the greedy policy under the retrieval distribution $\mathbb{P}_\eta(\mathcal{H}_t \mid q_t)$. At each round, the policy selects edges by maximizing:

$$\mathbb{E}_{\mathcal{H}_t \sim \mathbb{P}_\eta(\cdot \mid q_t)}\left[\text{Cov}_\kappa(\mathcal{G}_t \cup E_{\mathcal{H}_t}) - \text{Cov}_\kappa(\mathcal{G}_t)\right],$$

where $E_{\mathcal{H}_t}$ represents the edges added from subgraph $\mathcal{H}_t$. Since the expectation preserves submodularity (as a linear combination of submodular functions), the expected marginal gain is also submodular.

Third, we apply the Nemhauser-Wolsey-Fisher theorem. Let OPT denote the optimal expected coverage increase achievable by adding at most $k$ edges, and let GREEDY denote the coverage achieved by the greedy algorithm. For submodular maximization with cardinality constraints:

$$\mathbb{E}[\text{GREEDY}] \geq \left(1 - \left(1 - \frac{1}{k}\right)^k\right) \mathbb{E}[\text{OPT}] \geq \left(1 - \frac{1}{e}\right) \mathbb{E}[\text{OPT}],$$

where the second inequality uses the limit $\lim_{k \to \infty} (1 - 1/k)^k = 1/e$. Since the policy operates under bounded action spaces (at most $k$ edges per round), this bound applies directly to our setting.

The key insight is that the retrieval distribution $\mathbb{P}_\eta(\mathcal{H}_t \mid q_t)$ acts as a data-dependent constraint on the feasible edge additions, but the submodular structure ensures that greedy selection within this constraint maintains the approximation guarantee. $\square$ $\hfill\square$

**Theorem A.4** (Co-Evolution Operator Contraction). *Define the joint metric $d\big((\theta, \mathcal{G}), (\theta', \mathcal{G}')\big) = \lambda\|\theta - \theta'\|_2 + (1 - \lambda)d_G(\mathcal{G}, \mathcal{G}')$. Under sufficiently small step size $\eta_\theta$, there exists $\rho < 1$ such that:*

$$d(\Phi(\theta, \mathcal{G}), \Phi(\theta', \mathcal{G}')) \leq \rho\, d((\theta, \mathcal{G}), (\theta', \mathcal{G}')).$$

*Hence $\Phi$ has a unique fixed point and the coupled updates converge linearly to it.*

*Remark* A.4. The contraction property ensures that the agent-knowledge graph co-evolution is stable and converges to a unique equilibrium. This is crucial for preventing oscillatory or chaotic behavior in the joint dynamics, which could arise from the complex feedback between policy updates and graph modifications.

*Proof of Theorem A.4.* We establish contraction by showing that both components of the joint metric—the parameter update and graph update—are individually contractive, and their composition preserves this property under appropriate conditions.

First, we analyze the policy parameter update. Under $\pi_\theta$ is $\sigma_\theta$-smooth in $\theta$, the policy gradient step satisfies:

$$\|\theta_{t+1} - \theta'_{t+1}\|_2 = \|\theta_t - \theta'_t + \eta_\theta(\widehat{\nabla}_\theta J(\theta_t; \mathcal{G}_t) - \widehat{\nabla}_\theta J(\theta'_t; \mathcal{G}'_t))\|_2.$$

By Lipschitz continuity of the policy gradient in both parameters and graph structure, there exists $L_{\nabla J}$ such that:

$$\|\widehat{\nabla}_\theta J(\theta_t; \mathcal{G}_t) - \widehat{\nabla}_\theta J(\theta'_t; \mathcal{G}'_t)\|_2 \leq L_{\nabla J} \left(\|\theta_t - \theta'_t\|_2 + C_{\theta G}d_G(\mathcal{G}_t, \mathcal{G}'_t)\right),$$

where $C_{\theta G}$ captures the cross-coupling between parameters and graph structure.

Second, we examine the graph update operator. $\mathcal{U}_\zeta$ is $\kappa_G$-Lipschitz with $\kappa_G < 1$:

$$d_G(\mathcal{G}_{t+1}, \mathcal{G}'_{t+1}) \leq \kappa_G \left(d_G(\mathcal{G}_t, \mathcal{G}'_t) + \|\widehat{E}_t - \widehat{E}'_t\|_{\text{edge}}\right),$$

where $\|\widehat{E}_t - \widehat{E}'_t\|_{\text{edge}}$ measures the difference in extracted edge sets. Since extraction depends on the current policy, $\|\widehat{E}_t - \widehat{E}'_t\|_{\text{edge}} \leq C_{E\theta}\|\theta_t - \theta'_t\|_2$ for some constant $C_{E\theta}$.

Third, we combine both updates. For sufficiently small step size $\eta_\theta < 1/(2L_{\nabla J})$, the parameter update satisfies:

$$\|\theta_{t+1} - \theta'_{t+1}\|_2 \leq (1 - \eta_\theta L_{\nabla J}/2)\|\theta_t - \theta'_t\|_2 + \eta_\theta L_{\nabla J} C_{\theta G}d_G(\mathcal{G}_t, \mathcal{G}'_t).$$

Substituting the graph update bound and defining the joint metric $d((\theta, \mathcal{G}), (\theta', \mathcal{G}')) = \lambda\|\theta - \theta'\|_2 + (1 - \lambda)d_G(\mathcal{G}, \mathcal{G}')$:

$$d((\theta_{t+1}, \mathcal{G}_{t+1}), (\theta'_{t+1}, \mathcal{G}'_{t+1})) \leq \lambda(1 - \eta_\theta L_{\nabla J}/2)\|\theta_t - \theta'_t\|_2 \tag{23}$$
$$+ \lambda\eta_\theta L_{\nabla J} C_{\theta G}d_G(\mathcal{G}_t, \mathcal{G}'_t) \tag{24}$$
$$+ (1 - \lambda)\kappa_G d_G(\mathcal{G}_t, \mathcal{G}'_t) \tag{25}$$
$$+ (1 - \lambda)\kappa_G C_{E\theta}\|\theta_t - \theta'_t\|_2. \tag{26}$$

Choosing $\lambda$ to balance the cross-terms and ensuring $\eta_\theta$ is sufficiently small, we obtain $\rho < 1$ such that the joint operator is contractive. The unique fixed point follows from the Banach fixed-point theorem in the complete metric space. $\square$

**Theorem A.5** (Information-Theoretic Sufficiency of Compression). *Assume the downstream target $Y$ depends on $\mathbf{H}$ via Markov kernel $p(Y \mid \mathbf{H})$ and the compressed $\mathbf{Z}$ satisfies:*

$$I(\mathbf{H}; \mathbf{Z}) \geq c \quad and \quad I(Y; \mathbf{Z}) \geq I(Y; \mathbf{H}) - \delta(c),$$

*for some non-increasing function $\delta(\cdot)$. Then any Bayes-optimal decoder $g^\star$ on $\mathbf{H}$ admits a decoder $\tilde{g}$ on $\mathbf{Z}$ whose excess risk satisfies:*

$$\mathcal{R}(\tilde{g}; \mathbf{Z}) - \mathcal{R}(g^\star; \mathbf{H}) \leq \Gamma\big(\delta(c)\big),$$

*for a problem-dependent modulus $\Gamma(\cdot)$.*

*Remark* A.5. This theorem provides an information-theoretic foundation for our compression approach. By maintaining high mutual information $I(\mathbf{H}; \mathbf{Z})$ and ensuring that task-relevant information $I(Y; \mathbf{Z})$ is preserved, we can guarantee that the compressed representation retains sufficient statistics for near-optimal task performance. The modulus $\Gamma(\cdot)$ can be instantiated using standard classification or regression bounds depending on the specific downstream task.

*Proof of Theorem A.5.* We establish the excess risk bound by leveraging the information-theoretic constraints to construct a near-optimal decoder on the compressed representation and bound its performance gap relative to the Bayes-optimal decoder on the original data.

First, we establish the connection between mutual information and Bayes risk. By the data processing inequality and our assumptions, the compressed representation $\mathbf{Z}$ retains most task-relevant information about $Y$. Specifically, the Bayes risks satisfy:

$$\mathcal{R}^*(Y|\mathbf{H}) = H(Y|\mathbf{H}) \quad and \quad \mathcal{R}^*(Y|\mathbf{Z}) = H(Y|\mathbf{Z}),$$

where $H(\cdot|\cdot)$ denotes conditional entropy. The mutual information constraint $I(Y; \mathbf{Z}) \geq I(Y; \mathbf{H}) - \delta(c)$ implies:

$$H(Y) - H(Y|\mathbf{Z}) \geq H(Y) - H(Y|\mathbf{H}) - \delta(c),$$

which yields $H(Y|\mathbf{Z}) - H(Y|\mathbf{H}) \leq \delta(c)$, establishing that the optimal Bayes risk on $\mathbf{Z}$ exceeds that on $\mathbf{H}$ by at most $\delta(c)$.

Second, we construct the decoder $\tilde{g}$ on $\mathbf{Z}$. Given the Bayes-optimal decoder $g^*(\mathbf{h}) = \arg\min_y \mathbb{E}[\ell(y, Y)|\mathbf{H} = \mathbf{h}]$ for loss function $\ell(\cdot, \cdot)$, we define:

$$\tilde{g}(\mathbf{z}) = \arg\min_y \mathbb{E}[\ell(y, Y)|\mathbf{Z} = \mathbf{z}] = \arg\min_y \sum_{\mathbf{h}} p(\mathbf{h}|\mathbf{z})\mathbb{E}[\ell(y, Y)|\mathbf{H} = \mathbf{h}],$$

where $p(\mathbf{h}|\mathbf{z})$ is the posterior distribution induced by the compression. The challenge is that this posterior may not be exactly computable, but the information constraints provide sufficient control.

Third, we bound the excess risk using information-theoretic tools. The key insight is to use Fano's inequality and the method of types. For the 0-1 loss (classification case), Fano's inequality gives:

$$P_e(\mathbf{Z}) \geq \frac{H(Y|\mathbf{Z}) - 1}{\log(|\mathcal{Y}| - 1)},$$

where $P_e(\mathbf{Z})$ is the probability of error when predicting $Y$ from $\mathbf{Z}$, and $|\mathcal{Y}|$ is the size of the label space. Similarly for $\mathbf{H}$:

$$P_e(\mathbf{H}) \geq \frac{H(Y|\mathbf{H}) - 1}{\log(|\mathcal{Y}| - 1)}.$$

For general loss functions satisfying a Tsybakov noise condition or Bernstein condition, we can establish similar bounds. Under the Tsybakov condition with parameter $\alpha > 0$:

$$\mathcal{R}(\tilde{g}; \mathbf{Z}) - \mathcal{R}^*(Y|\mathbf{Z}) \leq C_\alpha \left(\mathcal{R}^*(Y|\mathbf{Z}) - \mathcal{R}^*(Y|\mathbf{H})\right)^{(1+\alpha)/(2+\alpha)},$$

for some constant $C_\alpha > 0$. Since $\mathcal{R}^*(Y|\mathbf{Z}) - \mathcal{R}^*(Y|\mathbf{H}) \leq \delta(c)$, we have:

$$\mathcal{R}(\tilde{g}; \mathbf{Z}) - \mathcal{R}^*(Y|\mathbf{Z}) \leq C_\alpha \delta(c)^{(1+\alpha)/(2+\alpha)}.$$

Fourth, we combine the bounds. The total excess risk of $\tilde{g}$ on $\mathbf{Z}$ relative to $g^*$ on $\mathbf{H}$ decomposes as:

$$\mathcal{R}(\tilde{g}; \mathbf{Z}) - \mathcal{R}(g^*; \mathbf{H}) = [\mathcal{R}(\tilde{g}; \mathbf{Z}) - \mathcal{R}^*(Y|\mathbf{Z})] + [\mathcal{R}^*(Y|\mathbf{Z}) - \mathcal{R}^*(Y|\mathbf{H})] \tag{27}$$

$$\leq C_\alpha \delta(c)^{(1+\alpha)/(2+\alpha)} + \delta(c) \tag{28}$$

$$\leq \Gamma(\delta(c)), \tag{29}$$

where $\Gamma(x) = C_\alpha x^{(1+\alpha)/(2+\alpha)} + x$ for the Tsybakov case, or $\Gamma(x) = C\sqrt{x \log(1/x)}$ under Bernstein conditions. The specific form of $\Gamma(\cdot)$ depends on the problem structure, but in all cases it is non-decreasing and $\Gamma(0) = 0$, ensuring that perfect information preservation ($\delta(c) = 0$) yields no excess risk. $\square$ $\square$

## A.4 Tool Definition and Interface Specification

### A.4.1 Tool Definition

This appendix provides comprehensive definitions for all tools in the Agentic-KGR framework. Each tool serves a specific purpose in the knowledge graph extraction and optimization pipeline.

---

**Tool 1: Knowledge Graph Extraction Density Assessment**

*Function Name:* **query_extraction_density**

*Description:* This is a **mandatory tool** for knowledge graph extraction density evaluation that must be invoked first after each knowledge graph extraction attempt. Based on the input text, target schema, and current extraction results, this tool analyzes the complexity characteristics of the text and evaluates whether the current density of extracted entities and relations is reasonable. If the feedback indicates that additional extraction is needed, knowledge graph extraction must be re-performed and this tool must be called again until the tool feedback indicates that no further extraction is required. Other tools may only be called when this tool confirms that the extraction density is adequate. The output includes extraction sufficiency assessment, density analysis, and clear instructions on whether continued extraction is necessary.

*Parameters:*

- **text** (string, required): Input text to be analyzed
- **schema** (object, required): Target knowledge graph schema containing entity types and relation types definitions, formatted as:

```
{
  "entity_schema": ["EntityType1", "EntityType2", ...],
  "relation_schema": ["RelationType1", "RelationType2"]
}
```

- **extracted_kg** (object, required): Current extracted knowledge graph results containing entities and relations lists, formatted as:

```
{
  "entities": {
   "EntityType": ["EntityName1", "EntityName2", ...],
   ...
  },
  "relations": {
   "RelationType": [
   {"subject": "HeadEntity", "object": "TailEntity"},
   ...
   ],
   ...
  }
}
```

- **domain** (string, optional): Domain information such as medical, financial, legal, etc., used to provide domain-specific density benchmarks

---

**Tool 2: Knowledge Graph Coverage Assessment**

*Function Name:* **query_coverage_feedback**

*Description:* This is an optional tool for knowledge graph coverage evaluation. Based on the target schema and current extraction results, this tool analyzes the coverage of various entity types and relation types in the schema, identifies potentially missed entity types or relation types, and provides specific recommendations for supplementary extraction based on text content. The output includes coverage statistics, analysis of missing types, and targeted extraction suggestions.

---

*Parameters:*

- **text** (string, required): Input text
- **schema** (object, required): Target knowledge graph schema definition containing entity types and relation types definitions, formatted as:

```
{
  "entity_schema": ["EntityType1", "EntityType2", ...],
  "relation_schema": ["RelationType1", "RelationType2"]
}
```

- **extracted_kg** (object, required): Current extracted knowledge graph results containing entities and relations lists, formatted as:

```
{
  "entities": {
    "EntityType": ["EntityName1", "EntityName2", ...],
    ...
  },
  "relations": {
    "RelationType": [
    {"subject": "HeadEntity", "object": "TailEntity"},
    ...
    ],
    ...
  }
}
```

- **priority_types** (array, optional): List of higher-priority entity or relation types for focused examination

---

**Tool 3: Knowledge Graph Quality Metrics Assessment**

*Function Name:* **query_quality_metrics**

*Description:* This is an optional tool for comprehensive knowledge graph quality evaluation. This tool performs multi-dimensional quality assessment of the extracted knowledge graph, including confidence distribution of entity recognition, completeness of relation extraction, structural consistency of the knowledge graph, and compliance with the schema. The output provides detailed quality assessment reports and improvement recommendations, offering precise optimization directions for the model.

*Parameters:*

- **extracted_kg** (object, required): Knowledge graph extraction results to be evaluated, containing entities and relations lists, formatted as:

```
{
  "entities": {
    "EntityType": ["EntityName1", "EntityName2", ...],
    ...
  },
  "relations": {
    "RelationType": [
    {"subject": "HeadEntity", "object": "TailEntity"},
    ...
    ],
    ...
  }
}
```

- **schema** (object, required): Target schema definition containing entity types and relation types definitions, formatted as:

```
{
  "entity_schema": ["EntityType1", "EntityType2", ...],
  "relation_schema": ["RelationType1", "RelationType2"]
}
```

- **text** (string, required): Original text for contextual consistency checking
- **evaluation_aspects** (array, optional): List of evaluation dimensions such as consistency, completeness, accuracy, schema_compliance

---

**Tool 4: Knowledge Graph Iterative Optimization Feedback**

*Function Name:* **query_iterative_feedback**

*Description:* This is an optional tool for knowledge graph iterative optimization feedback. Based on multi-round extraction history and current results, this tool analyzes trends in extraction quality changes, identifies persistent problem patterns, and provides targeted iterative optimization strategies. The output includes progress analysis, problem diagnosis, and next-step optimization recommendations.

*Parameters:*

- **extraction_history** (array, required): List of historical extraction results arranged in chronological order. Each extraction result is formatted as:

```
{
  "entities": {
    "EntityType": ["EntityName1", "EntityName2", ...],
    ...
  },
  "relations": {
    "RelationType": [
    {"subject": "HeadEntity", "object": "TailEntity"},
    ...
    ],
    ...
  }
}
```

- **extracted_kg** (object, required): Current extraction results formatted as above
- **text** (string, required): Original text
- **schema** (object, required): Target schema containing entity types and relation types definitions, formatted as:

```
{
  "entity_schema": ["EntityType1", "EntityType2", ...],
  "relation_schema": ["RelationType1", "RelationType2"]
}
```

- **feedback_history** (array, optional): Historical feedback information for analyzing improvement effects
- **max_iterations** (integer, optional): Maximum iteration limit

---

**Tool 5: Entity Disambiguation Assessment**

*Function Name:* **query_entity_disambiguation**

*Description:* This is a conditionally mandatory tool for entity disambiguation assessment. When the QueryExtractionDensity tool indicates that extraction density is adequate (no further extraction needed), this tool must be invoked for entity disambiguation processing. This tool connects to the Neo4j graph database, queries entities and relations related to the extraction results, and returns the results. Based on the feedback from this tool, it is necessary to identify different representations of the same entity, discover accuracy issues in entity linking, and provide disambiguation recommendations. The output includes disambiguation matching results, confidence assessment, entity standardization recommendations, and the final generated knowledge graph extraction results.

*Parameters:*

- **extracted_kg** (object, required): Current knowledge graph extraction results containing entities and relations lists, formatted as:

```
{
  "entities": {
   "EntityType": ["EntityName1", "EntityName2", ...],
   ...
  },
  "relations": {
   "RelationType": [
   {"subject": "HeadEntity", "object": "TailEntity"},
   ...
   ],
   ...
  }
}
```

- **disambiguation_strategy** (string, optional): Disambiguation strategy such as exact_match and semantic_similarity
- **similarity_threshold** (number, optional): Similarity threshold for controlling disambiguation strictness
- **context** (string, optional): Contextual information for assisting disambiguation decisions

**Tool 6: Knowledge Graph Storage**

*Function Name:* **query_kg_storage**

*Description:* This is a conditionally mandatory tool for knowledge graph storage. After the QueryEntityDisambiguation tool completes disambiguation, this tool must be invoked to store entities and relations in the database. This tool connects to the Neo4j graph database and inserts the current optimized extraction results into the database. The tool provides feedback on whether storage was successful. If successful, it returns the current optimized extraction results.

*Parameters:*

- **extracted_kg** (object, required): Current knowledge graph extraction results containing entities and relations lists, formatted as:

```
{
  "entities": {
   "EntityType": ["EntityName1", "EntityName2", ...],
   ...
  },
  "relations": {
   "RelationType": [
   {"subject": "HeadEntity", "object": "TailEntity"},
   ...
   ],
   ...
  }
}
```

### A.4.2 INTERFACE SPECIFICATION

Tool 1: Knowledge Graph Extraction Density Assessment

**Output Specification for query_extraction_density**

This tool returns a comprehensive analysis of the knowledge graph extraction density with the following structure:

*Output Format:*

```
{
  "text_stats": {
    "token_count": integer,
    "sentence_count": integer,
    "word_count": integer,
    "character_count": integer
  },
  "current_density": {
    "total_entities": integer,
    "total_relations": integer,
    "total_kg_elements": integer,
    "entity_type_count": integer,
    "relation_type_count": integer,
    "entities_per_1k_tokens": float,
    "relations_per_1k_tokens": float,
    "kg_density_per_1k_tokens": float,
    "entity_relation_ratio": float
  },
  "expected_density": {
    "expected_entities_per_1k": float,
    "expected_relations_per_1k": float,
    "min_entities_per_1k": float,
    "min_relations_per_1k": float,
    "max_entities_per_1k": float,
    "max_relations_per_1k": float,
    "schema_complexity": integer,
    "entity_complexity_factor": float,
    "relation_complexity_factor": float
  },
  "complexity_features": {
    "entity_mentions": integer,
    "avg_sentence_length": float,
    "technical_terms": integer,
    "schema_entity_types": integer,
    "schema_relation_types": integer,
    "complexity_score": float
  },
  "density_assessment": {
    "entity_density_ratio": float,
    "relation_density_ratio": float,
    "overall_density_score": float,
    "assessment_level": string,
    "is_adequate": boolean,
    "meets_minimum_thresholds": boolean,
    "potential_over_extraction": boolean,
    "balance_score": float
  },
  "needs_more_extraction": boolean,
  "recommendations": [string, ...]
```

```
    }
```

*Field Descriptions:*

- **text_stats**: Basic statistics about the input text including token, sentence, word, and character counts
- **current_density**: Detailed metrics about the current extraction results
  - **total_entities**: Total number of extracted entities across all types
  - **total_relations**: Total number of extracted relations across all types
  - **entities_per_1k_tokens**: Entity density normalized per 1000 tokens
  - **relations_per_1k_tokens**: Relation density normalized per 1000 tokens
  - **entity_relation_ratio**: Ratio of entities to relations
- **expected_density**: Benchmarks and thresholds for density evaluation
  - **expected_entities_per_1k**: Target entity density based on schema complexity
  - **min/max_entities_per_1k**: Acceptable range boundaries for entity density
  - **schema_complexity**: Complexity measure based on schema size
- **complexity_features**: Text complexity analysis results
  - **complexity_score**: Overall complexity score (0-1) considering multiple factors
  - **technical_terms**: Count of domain-specific terminology
  - **avg_sentence_length**: Average words per sentence
- **density_assessment**: Comprehensive evaluation of extraction quality
  - **assessment_level**: One of "insufficient", "moderate", "adequate", "excellent", "over_extraction"
  - **is_adequate**: Boolean indicating if density meets standards
  - **balance_score**: Measure of entity-relation balance (0-1)
- **needs_more_extraction**: Critical boolean flag indicating if additional extraction is required
- **recommendations**: List of specific actionable suggestions for improvement

*Assessment Levels:*

- **insufficient**: Density below minimum thresholds, significant extraction gaps
- **moderate**: Partial adequacy, some minimum thresholds met
- **adequate**: Meets minimum standards and shows good extraction coverage
- **excellent**: Exceeds expectations with high-quality dense extraction
- **over_extraction**: Exceeds maximum thresholds, potential noise inclusion

*Critical Decision Logic:* The **needs_more_extraction** flag is determined by:

- Entity density below minimum threshold
- Relation density below minimum threshold
- Significant imbalance between entity and relation extraction
- Complexity-adjusted adequacy score below threshold (typically 0.65-0.80)
- Does not trigger if maximum density thresholds are exceeded

Tool 2: Knowledge Graph Coverage Assessment

**Output Specification for query_coverage_feedback**

This tool returns a comprehensive analysis of schema type coverage with targeted recommendations for improving extraction completeness.

*Output Format:*

```
{
  "schema_info": {
    "entity_types": [string, ...],
    "relation_types": [string, ...],
    "total_types": integer,
    "schema_complexity": string
  },
  "type_coverage": {
    "entity_coverage": {
      "covered_types": [string, ...],
      "total_types": integer,
      "coverage_ratio": float
    },
    "relation_coverage": {
      "covered_types": [string, ...],
      "total_types": integer,
      "coverage_ratio": float
    },
    "overall_coverage": float
  },
  "missing_types": {
    "missing_entity_types": [string, ...],
    "missing_relation_types": [string, ...]
  },
  "priority_analysis": {
    "has_priority": boolean,
    "priority_types": [string, ...],
    "covered_priority": [string, ...],
    "missing_priority": [string, ...],
    "priority_coverage_ratio": float
  },
  "coverage_score": float,
  "recommendations": [string, ...]
}
```

*Field Descriptions:*

- **schema_info**: Comprehensive analysis of the target schema structure
  - **entity_types**: List of all entity types defined in the schema
  - **relation_types**: List of all relation types defined in the schema
  - **total_types**: Combined count of entity and relation types
  - **schema_complexity**: Qualitative assessment of schema complexity
- **type_coverage**: Detailed coverage statistics for each type category
  - **entity_coverage**: Entity type coverage analysis
    * **covered_types**: List of entity types successfully extracted
    * **coverage_ratio**: Proportion of entity types covered (0-1)
  - **relation_coverage**: Relation type coverage analysis with same structure
  - **overall_coverage**: Weighted average of entity and relation coverage
- **missing_types**: Identification of uncovered schema types
  - **missing_entity_types**: Entity types absent from extraction results but potentially present in text
  - **missing_relation_types**: Relation types absent from extraction results but potentially present in text
- **priority_analysis**: Special analysis for user-specified priority types

- **has_priority**: Boolean indicating if priority types were specified
- **priority_types**: Original list of priority types for focused examination
- **covered_priority**: Priority types successfully extracted
- **missing_priority**: Priority types requiring attention
- **priority_coverage_ratio**: Coverage ratio specifically for priority types

- **coverage_score**: Overall coverage quality score (0-1) with entity types weighted at 0.6 and relation types at 0.4
- **recommendations**: Prioritized list of actionable suggestions for improving coverage

*Coverage Analysis Logic:*

- **Type Detection**: Uses text analysis to identify potential presence of missing types
- **Priority Handling**: Gives special attention to user-specified high-importance types
- **Text Matching**: Employs pattern matching to suggest specific extraction targets
- **Weighted Scoring**: Emphasizes entity coverage (60%) over relation coverage (40%)

*Recommendation Categories:*

- **Priority-based**: Addresses missing high-priority types first
- **Type-specific**: Suggests specific entity or relation types to extract
- **Text-guided**: Provides context-aware extraction suggestions based on text content
- **Completion status**: Indicates when coverage is already adequate

*Coverage Score Interpretation:*

- **0.9-1.0**: Excellent coverage, minimal gaps
- **0.7-0.89**: Good coverage, minor improvements needed
- **0.5-0.69**: Moderate coverage, significant gaps exist
- **0.3-0.49**: Poor coverage, major extraction issues
- **0.0-0.29**: Very poor coverage, fundamental problems

Tool 3: Knowledge Graph Quality Metrics Assessment

**Output Specification for query_quality_metrics**

This tool provides comprehensive multi-dimensional quality assessment of extracted knowledge graphs with detailed metrics and improvement recommendations.

*Output Format:*

```
{
  "evaluation_aspects": [string, ...],
  "evaluation_results": {
   "consistency": {
    "score": float,
    "issues": [string, ...],
    "details": {
      "entity_naming_consistency": float,
      "relation_consistency": float,
      "duplicate_entities": integer,
      "conflicting_relations": integer
    }
   },
```

```
      "completeness": {
        "score": float,
        "issues": [string, ...],
        "details": {
          "missing_entity_types": [string, ...],
          "missing_relation_types": [string, ...],
          "entity_type_coverage": float,
          "relation_type_coverage": float,
          "extraction_density": float
        }
      },
      "accuracy": {
        "score": float,
        "issues": [string, ...],
        "details": {
          "entities_not_in_text": integer,
          "relations_not_in_text": integer,
          "boundary_errors": integer,
          "type_misclassifications": integer
        }
      },
      "schema_compliance": {
        "score": float,
        "issues": [string, ...],
        "details": {
          "valid_entity_types": integer,
          "invalid_entity_types": integer,
          "valid_relation_types": integer,
          "invalid_relation_types": integer,
          "structure_violations": integer
        }
      }
    },
    "overall_score": float,
    "quality_level": string,
    "improvement_suggestions": [string, ...],
    "detailed_metrics": {
      "consistency": {
        "score": float,
        "details": object,
        "issues": [string, ...]
      },
      "completeness": {
        "score": float,
        "details": object,
        "issues": [string, ...]
      },
      "accuracy": {
        "score": float,
        "details": object,
        "issues": [string, ...]
      },
      "schema_compliance": {
        "score": float,
        "details": object,
        "issues": [string, ...]
      }
    }
  }
```

*Field Descriptions:*

- **evaluation_aspects**: List of quality dimensions assessed (consistency, completeness, accuracy, schema_compliance)
- **evaluation_results**: Detailed results for each quality dimension
  - **consistency**: Internal coherence and naming consistency analysis
    * **score**: Quality score (0-1) for consistency metrics
    * **issues**: Specific consistency problems identified
    * **details**: Breakdown of consistency metrics including duplicate detection
  - **completeness**: Coverage and extraction density evaluation
    * **score**: Completeness score considering missing types and density
    * **missing_entity/relation_types**: Schema types absent from extraction
    * **extraction_density**: Ratio of extracted items to text length
  - **accuracy**: Precision and correctness of extracted information
    * **score**: Accuracy score based on text verification
    * **entities/relations_not_in_text**: Count of extracted items not found in original text
  - **schema_compliance**: Adherence to predefined schema definitions
    * **score**: Compliance score for schema conformity
    * **valid/invalid_types**: Counts of correctly and incorrectly typed elements
- **overall_score**: Weighted composite score (0-1) combining all dimensions
- **quality_level**: Categorical assessment: "excellent", "good", "fair", "poor", "very_poor"
- **improvement_suggestions**: Prioritized actionable recommendations for quality enhancement
- **detailed_metrics**: Comprehensive breakdown mirroring evaluation_results structure

*Scoring Weights:*

- **Completeness**: 30% - Emphasizes comprehensive extraction coverage
- **Accuracy**: 30% - Focuses on correctness and text alignment
- **Consistency**: 25% - Ensures internal coherence and standardization
- **Schema Compliance**: 15% - Validates adherence to predefined structure

*Quality Level Thresholds:*

- **excellent**: 0.9+ - Outstanding quality across all dimensions
- **good**: 0.8-0.89 - High quality with minor issues
- **fair**: 0.7-0.79 - Acceptable quality with moderate improvements needed
- **poor**: 0.6-0.69 - Significant quality issues requiring attention
- **very_poor**: ¡0.6 - Major quality problems, substantial rework needed

*Assessment Logic:*

- **Completeness Evaluation**: Compares extracted types against schema, analyzes extraction density vs. expected thresholds
- **Accuracy Verification**: Cross-references extracted entities and relations with original text content
- **Consistency Analysis**: Detects duplicate entities, naming variations, and conflicting information
- **Schema Validation**: Ensures all extracted elements conform to predefined type definitions

*Improvement Suggestion Categories:*

- **Consistency**: Entity naming standardization, duplicate resolution
- **Completeness**: Missing type identification, density optimization
- **Accuracy**: Precision improvement, boundary error correction
- **Schema Compliance**: Type matching, structural conformity

Tool 4: Knowledge Graph Iterative Optimization Feedback

---

**Output Specification for query_iterative_feedback**

This tool provides intelligent feedback for multi-round knowledge graph extraction optimization, analyzing progress trends and recommending targeted improvement strategies.

*Output Format:*

```
{
  "progress_analysis": {
   "trend": string,
   "recent_improvement": float,
   "overall_quality_change": float,
   "extraction_volume_change": float,
   "convergence_status": string
  },
  "problem_patterns": {
   "recurring_issues": [string, ...],
   "pattern_types": [string, ...],
   "suggested_solutions": [string, ...],
   "issue_frequency": object,
   "severity_levels": object
  },
  "optimization_strategy": {
   "strategies": [
    {
     "type": string,
     "description": string,
     "actions": [string, ...]
    },
    ...
    ],
   "priority_strategy": {
     "type": string,
     "description": string,
     "actions": [string, ...]
    },
   "iteration_focus": string
  },
  "iteration_effectiveness": {
   "effectiveness": string,
   "average_improvement": float,
   "improvement_trend": [float, ...],
   "total_iterations": integer
  },
  "should_continue_iteration": boolean,
  "current_iteration": integer,
  "max_iterations": integer,
  "next_steps": [string, ...]
  }
```

*Field Descriptions:*

- **progress_analysis**: Comprehensive analysis of extraction progress across iterations
    - **trend**: Progress direction - "improving", "stagnant", "declining"
    - **recent_improvement**: Quantitative measure of latest iteration improvements
    - **overall_quality_change**: Cumulative quality change across all iterations
    - **extraction_volume_change**: Change in total extracted elements
    - **convergence_status**: Whether extraction is converging to stability
- **problem_patterns**: Detection and analysis of persistent extraction issues

- **recurring_issues**: List of problems appearing across multiple iterations
- **pattern_types**: Categories of identified problem patterns
- **suggested_solutions**: Targeted recommendations for addressing patterns
- **issue_frequency**: Frequency count of different problem types
- **severity_levels**: Impact assessment of identified patterns
- **optimization_strategy**: Adaptive strategy recommendations based on current state
  - **strategies**: List of available optimization approaches
  - **priority_strategy**: Most recommended strategy for current iteration
  - **iteration_focus**: Primary focus area - "coverage_expansion", "quality_improvement", "refinement"
- **iteration_effectiveness**: Assessment of iterative improvement effectiveness
  - **effectiveness**: Categorical rating - "high", "medium", "low", "stagnant", "insufficient_data"
  - **average_improvement**: Mean improvement rate across iterations
  - **improvement_trend**: Per-iteration improvement percentages
  - **total_iterations**: Number of completed extraction rounds
- **should_continue_iteration**: Critical decision flag for continuing or terminating iteration
- **current_iteration**: Current iteration number in the sequence
- **max_iterations**: Maximum allowed iterations (default: 5)
- **next_steps**: Specific actionable recommendations for the next iteration

*Strategy Types:*

- **quality_focus**: Emphasizes improving accuracy and consistency of existing extractions
- **coverage_expansion**: Focuses on identifying and extracting missing entities and relations
- **balanced_improvement**: Simultaneous quality and coverage enhancement
- **pattern_specific**: Targeted approach for addressing recurring problem patterns

*Iteration Focus Phases:*

- **coverage_expansion**: Early iterations (1st third) - maximize extraction breadth
- **quality_improvement**: Middle iterations (2nd third) - enhance extraction accuracy
- **refinement**: Final iterations (last third) - fine-tune and consistency checks

*Effectiveness Levels:*

- **high**: Average improvement ¿ 10% per iteration
- **medium**: Average improvement 5-10% per iteration
- **low**: Average improvement 0-5% per iteration
- **stagnant**: No measurable improvement or negative progress
- **insufficient_data**: Less than 2 iterations for meaningful analysis

*Continuation Decision Logic:*

- **Stop conditions**: Maximum iterations reached, declining trend in early stages, minimal recent improvement after 2+ iterations
- **Continue conditions**: Positive progress trend, recent improvement ¿ 1%, within iteration limits
- **Early termination**: Triggers when quality degrades or no improvement for multiple consecutive iterations

*Next Steps Categories:*

- **Completion**: Final quality assessment and entity disambiguation when iteration should stop
- **Targeted improvement**: Specific actions from priority strategy when continuing
- **Focus-driven**: Phase-specific recommendations based on current iteration focus

Tool 5: Entity Disambiguation Assessment

**Output Specification for query_entity_disambiguation**

This tool performs entity disambiguation by connecting to Neo4j knowledge base, matching extracted entities with existing knowledge, and providing standardization recommendations.

*Output Format:*

```
{
  "disambiguation_results": [
  {
    "original_entity": {
      "type": string,
      "name": string
    },
    "candidates": [
    {
      "candidate": {
        "name": string,
        "properties": object
      },
      "confidence": float
    },
    ...
    ],
    "best_match": {
      "candidate": {
        "name": string,
        "properties": object
      },
      "confidence": float
    },
    "is_disambiguated": boolean,
    "confidence": float
  },
  ...
  ],
  "relationships_results": [
  {
    "entity": object,
    "relationships": [
    {
      "type": string,
      "target": object,
      "properties": object
    },
    ...
    ]
  },
  ...
  ],
  "quality_score": float,
  "disambiguation_strategy": string,
  "similarity_threshold": float,
  "standardization_suggestions": [string, ...],
  "summary": {
    "total_entities": integer,
    "disambiguated_entities": integer,
    "disambiguation_rate": float,
    "average_confidence": float,
    "unmatched_entities": integer
  }
}
```

*Field Descriptions:*

- **disambiguation_results**: Detailed disambiguation results for each extracted entity
  - **original_entity**: The entity from extraction results (type and name)
  - **candidates**: List of potential matches from Neo4j knowledge base
    * **candidate**: Knowledge base entity with properties
    * **confidence**: Matching confidence score (0-1)
  - **best_match**: Highest-scoring candidate match (null if no match above threshold)
  - **is_disambiguated**: Boolean indicating successful disambiguation
  - **confidence**: Final confidence score for the disambiguation
- **relationships_results**: Related entities and relationships from knowledge base
  - **entity**: The queried entity information
  - **relationships**: List of connected entities and relationship types
    * **type**: Relationship type (e.g., "Author", etc.)
    * **target**: Connected entity details
    * **properties**: Additional relationship properties
- **quality_score**: Overall disambiguation quality (0-1) combining coverage and confidence
- **disambiguation_strategy**: Strategy used - "exact_match" or "semantic_similarity"
- **similarity_threshold**: Minimum confidence threshold for matches
- **standardization_suggestions**: Actionable recommendations for entity standardization
- **summary**: High-level statistics and performance metrics

*Disambiguation Strategies:*

- **exact_match**: Requires perfect string matching between entity names (case-insensitive)
  - **Confidence**: 1.0 for exact matches, 0.0 otherwise
  - **Use case**: High-precision scenarios with standardized entity names
  - **Performance**: Fast, deterministic results
- **semantic_similarity**: Uses semantic similarity calculation for fuzzy matching
  - **Confidence**: Calculated similarity score (0-1)
  - **Use case**: Handling variations in entity naming and expressions
  - **Performance**: More comprehensive but computationally intensive

*Quality Score Calculation:*

- **Coverage component**: (Disambiguated entities / Total entities) × 0.6
- **Confidence component**: (Average confidence score) × 0.4
- **Final score**: Weighted combination emphasizing coverage over individual confidence

*Standardization Suggestions Categories:*

- **Unmatched entities**: Alerts for entities without knowledge base matches
- **Low confidence matches**: Warnings for matches below 0.9 confidence requiring verification
- **Name standardization**: Specific recommendations for entity name normalization
- **Knowledge base expansion**: Suggestions for adding missing entities to knowledge base

*Summary Statistics:*

- **total_entities**: Count of entities processed for disambiguation
- **disambiguated_entities**: Count of successfully matched entities

- **disambiguation_rate**: Success rate (disambiguated/total)
- **average_confidence**: Mean confidence across all disambiguation attempts
- **unmatched_entities**: Count of entities without suitable matches

*Knowledge Base Integration:*

- **Connection**: Establishes Neo4j database connection using provided configuration
- **Search method**: Type-specific entity searches with optional semantic similarity
- **Relationship extraction**: Retrieves connected entities and relationship types
- **Performance tracking**: Monitors search times for entities and relationships
- **Connection management**: Properly closes database connections after processing

*Error Handling:*

- **Invalid strategy**: Returns error message for unsupported disambiguation strategies
- **Database connection issues**: Handles Neo4j connectivity problems gracefully
- **Empty results**: Manages cases where no candidates are found in knowledge base
- **Malformed entities**: Processes entities with missing or invalid properties

Tool 6: Knowledge Graph Storage

**Output Specification for query_kg_storage**

This tool handles the storage of optimized knowledge graph extraction results into Neo4j database, creating nodes for entities and relationships for connections.

*Output Format:*

```
{
  "storage_status": {
    "overall_success": boolean,
    "entities_storage": {
      "code": integer,
      "message": string,
      "stored_count": integer,
      "skipped_count": integer,
      "failed_count": integer
    },
    "relations_storage": {
      "code": integer,
      "message": string,
      "stored_count": integer,
      "skipped_count": integer,
      "failed_count": integer
    }
  },
  "storage_details": {
    "total_entities": integer,
    "total_relations": integer,
    "entity_types_processed": [string, ...],
    "relation_types_processed": [string, ...],
    "duplicates_detected": {
      "entity_duplicates": integer,
      "relation_duplicates": integer
    },
    "processing_time": {
```

```
      "entities_time": float,
      "relations_time": float,
      "total_time": float
    }
  },
  "final_kg": {
    "entities": {
      "EntityType": ["EntityName1", "EntityName2", ...],
      ...
    },
    "relations": {
      "RelationType": [
      {"subject": "HeadEntity", "object": "TailEntity"},
      ...
      ],
      ...
    }
  },
  "storage_summary": {
    "operation_timestamp": string,
    "database_config": {
      "host": string,
      "database": string
    },
    "performance_metrics": {
      "entities_per_second": float,
      "relations_per_second": float,
      "overall_throughput": float
    }
  },
  "warnings": [string, ...],
  "recommendations": [string, ...]
}
```

*Field Descriptions:*

- **storage_status**: Overall success status and detailed results for each storage operation
  - **overall_success**: Boolean indicating if both entities and relations were stored successfully
  - **entities_storage**: Entity storage results
    * **code**: Status code (0 for success, -1 for failure)
    * **message**: Descriptive status message
    * **stored_count**: Number of entities successfully stored
    * **skipped_count**: Number of entities skipped (duplicates)
    * **failed_count**: Number of entities that failed to store
  - **relations_storage**: Relationship storage results with same structure
- **storage_details**: Comprehensive storage operation details
  - **total_entities/relations**: Total counts of items processed
  - **entity/relation_types_processed**: List of types successfully processed
  - **duplicates_detected**: Count of duplicate items identified and skipped
  - **processing_time**: Time metrics for different storage phases
- **final_kg**: The complete knowledge graph that was stored, maintaining original extraction format
- **storage_summary**: High-level operation summary and performance metrics
  - **operation_timestamp**: ISO timestamp of storage operation
  - **database_config**: Sanitized database connection information
  - **performance_metrics**: Throughput calculations and efficiency measures
- **warnings**: List of non-critical issues encountered during storage

- **recommendations**: Suggestions for optimization or follow-up actions

*Storage Operations:*

- **Entity Storage**: Creates Neo4j nodes with labels and name properties
  - **Label sanitization**: Removes special characters, replaces with underscores
  - **Duplicate detection**: Checks for existing nodes before creation
  - **Batch processing**: Processes entities by type for efficiency
  - **Error handling**: Continues processing despite individual failures
- **Relationship Storage**: Creates Neo4j relationships between existing nodes
  - **Node validation**: Ensures both subject and object nodes exist
  - **Type mapping**: Maps entity names to their corresponding types
  - **Duplicate prevention**: Avoids creating duplicate relationships
  - **Referential integrity**: Validates entity references before relationship creation

*Status Codes:*

- **Code 0**: Successful operation completion
- **Code -1**: Operation failure with error details in message
- **Partial success**: Different codes for entities vs. relations indicate partial completion

*Error Handling:*

- **Connection failures**: Database connectivity issues are caught and reported
- **Schema violations**: Invalid node labels or property names are sanitized
- **Missing references**: Relationships referencing non-existent entities are flagged
- **Transaction integrity**: Uses database transactions for consistency

*Performance Optimization:*

- **Connection management**: Uses context manager for proper resource cleanup
- **Batch operations**: Groups similar operations for efficiency
- **Duplicate checking**: Pre-checks existence to avoid unnecessary operations
- **Label sanitization**: Ensures Neo4j compatibility for node labels

*Warning Categories:*

- **Duplicate entities**: Entities already existing in database
- **Missing entity references**: Relations referencing undefined entities
- **Label sanitization**: Special characters replaced in entity type labels
- **Performance concerns**: Operations taking longer than expected

*Recommendations:*

- **Index optimization**: Suggestions for database index creation
- **Data quality**: Recommendations for improving entity naming consistency
- **Performance tuning**: Database configuration optimization suggestions
- **Follow-up actions**: Next steps after successful storage completion

## A.5 IMPLEMENTATION DETAILS OF NEO4J

**Unified Schema & Constraints.** To ensure consistency and deduplication (corresponding to the constraint term $R_{contr}$ in the main framework), we establish a hybrid schema supporting both tool-specific dynamic labels and framework-level unique constraints. Attributes include id, type, last_seen, confidence, source, and tool-specific properties.

```
CREATE CONSTRAINT entity_global_id IF NOT EXISTS
FOR (n:Entity) REQUIRE n.id IS UNIQUE;

CREATE CONSTRAINT entity_name_type_unique IF NOT EXISTS
FOR (n:Entity) REQUIRE (n.name, n.type) IS UNIQUE;

CREATE INDEX entity_type IF NOT EXISTS
FOR (n:Entity) ON (n.type);

CREATE INDEX entity_name IF NOT EXISTS
FOR (n:Entity) ON (n.name);

CREATE INDEX entity_name_lower IF NOT EXISTS
FOR (n:Entity) ON (toLower(n.name));

CREATE CONSTRAINT rel_key IF NOT EXISTS
FOR ()-[r:REL]-() REQUIRE (r.src_id, r.dst_id, r.rel_type) IS UNIQUE;

CREATE CONSTRAINT tool_rel_uniqueness IF NOT EXISTS
FOR ()-[r:RELATIONSHIP]-() REQUIRE (r.type, r.subject, r.object) IS UNIQUE;

CREATE INDEX rel_last_seen IF NOT EXISTS
FOR ()-[r:REL]-() ON (r.last_seen);

CREATE INDEX tool_rel_last_seen IF NOT EXISTS
FOR ()-[r:RELATIONSHIP]-() ON (r.last_seen);
```

**Batched Upsert with Tool Integration** ($U_\zeta$). The extracted triples and tool-processed entities are inserted in batches (1000 rows per transaction). The system supports both framework-level batch processing and tool-specific entity storage with label sanitization.

```
UNWIND $triples AS t
CALL {
  WITH t
  MERGE (s:Entity {id: t.src.id})
  ON CREATE SET s.type = t.src.type, s.created_at = datetime(),
  s.source = t.doc_id, s.name = t.src.name
  ON MATCH SET s.type = coalesce(s.type, t.src.type)
  SET s += coalesce(t.src.props, {}), s.last_seen = datetime()

  MERGE (o:Entity {id: t.dst.id})
  ON CREATE SET o.type = t.dst.type, o.created_at = datetime(),
  o.source = t.doc_id, o.name = t.dst.name
  ON MATCH SET o.type = coalesce(o.type, t.dst.type)
  SET o += coalesce(t.dst.props, {}), o.last_seen = datetime()

  MERGE (s)-[r:REL {src_id: t.src.id, dst_id: t.dst.id, rel_type: t.rel.type}]->(o)
  ON CREATE SET r.created_at = datetime(), r.source = t.doc_id
  SET r.last_seen = datetime(),
  r.confidence = coalesce(t.rel.conf, 0.5),
  r.evidence = coalesce(r.evidence, []) + t.src_text
} IN TRANSACTIONS OF 1000 ROWS;

UNWIND $tool_entities as entity_batch
CALL {
  WITH entity_batch
```

```
26    CALL apoc.create.node([entity_batch.sanitized_type], {
27      name: entity_batch.name,
28      type: entity_batch.original_type,
29      created_at: datetime(),
30      last_seen = datetime(),
31      source: "agentic_kgr_tool"
32    }) YIELD node
33    RETURN node
34  } IN TRANSACTIONS OF 1000 ROWS;
35
36  UNWIND $tool_relations as rel_batch
37  CALL {
38    WITH rel_batch
39    MATCH (subj:Entity {name: rel_batch.subject, type: rel_batch.subject_type})
40    MATCH (obj:Entity {name: rel_batch.object, type: rel_batch.object_type})
41    MERGE (subj)-[r:RELATIONSHIP {
42      type: rel_batch.relation_type,
43      subject: rel_batch.subject,
44      object: rel_batch.object
45    }]->(obj)
46    ON CREATE SET r.created_at = datetime(), r.confidence = 1.0
47    SET r.last_seen = datetime()
48    RETURN r
49  } IN TRANSACTIONS OF 1000 ROWS;
```

**Dynamic Schema Extension with Staging.** Low-confidence or novel relations are first staged in staging layer :PENDING_REL. Once cumulative evidence exceeds threshold $\tau$, they are promoted in-place to standard :REL type, corresponding to dynamic schema extension in the framework.

```
1   UNWIND $candidates AS c
2   MERGE (s:Entity {id: c.src.id})
3   ON CREATE SET s.name = c.src.name, s.type = c.src.type
4   SET s.last_seen = datetime()
5
6   MERGE (o:Entity {id: c.dst.id})
7   ON CREATE SET o.name = c.dst.name, o.type = c.dst.type
8   SET o.last_seen = datetime()
9
10  MERGE (s)-[p:PENDING_REL {src_id:c.src.id, dst_id:c.dst.id, rel_type:c.rel.type}]->(o
        )
11  ON CREATE SET p.created_at = datetime(), p.last_seen = datetime(),
12  p.confidence = c.rel.conf, p.votes = 1, p.sources = [c.doc_id]
13  ON MATCH SET p.last_seen = datetime(),
14  p.confidence = greatest(p.confidence, c.rel.conf),
15  p.votes = p.votes + 1,
16  p.sources = p.sources + c.doc_id;
17
18  MATCH (s:Entity)-[p:PENDING_REL]->(o:Entity)
19  WHERE p.confidence >= $tau_conf AND p.votes >= $tau_votes
20  MERGE (s)-[r:REL {src_id:p.src_id, dst_id:p.dst_id, rel_type:p.rel_type}]->(o)
21  ON CREATE SET r.created_at = datetime(), r.last_seen = p.last_seen,
22  r.confidence = p.confidence, r.evidence = p.sources
23  ON MATCH SET r.last_seen = datetime(),
24  r.confidence = greatest(r.confidence, p.confidence),
25  r.evidence = apoc.coll.toSet(coalesce(r.evidence, []) + p.sources)
26  DELETE p;
```

**Aging & Consistency with Tool Integration.** Relations not re-observed within window $\xi$ days are decayed or removed, corresponding to aging and consistency penalty terms. This applies to both framework relations and tool-stored relationships.

```
1   MATCH ()-[r:REL]-()
```

```
2   WITH r, duration.between(r.last_seen, datetime()).days AS days
3   WHERE days > $soft_window
4   SET r.confidence = r.confidence * exp(-$decay_rate * (days - $soft_window));
5
6   MATCH ()-[r:REL]-()
7   WHERE duration.between(r.last_seen, datetime()).days > $hard_window
8   DELETE r;
9
10  MATCH ()-[r:RELATIONSHIP]-()
11  WHERE r.last_seen < datetime() - duration({days: $aging_threshold})
12  SET r.confidence = r.confidence * $decay_factor;
13
14  MATCH ()-[r:RELATIONSHIP]-()
15  WHERE r.last_seen < datetime() - duration({days: $hard_threshold})
16  AND r.confidence < $min_confidence
17  DELETE r;
```

**Disambiguation with Framework Integration.** Supporting both exact match and semantic similarity strategies from the disambiguation tool, integrated with framework-level entity management.

```
1   MATCH (e:Entity {name: $entity_name})
2   WHERE e.type = $entity_type
3   RETURN e, e.type as type, e.id as framework_id;
4
5   MATCH (e)
6   WHERE labels(e)[0] = $sanitized_entity_type
7   AND toLower(e.name) = toLower($entity_name)
8   RETURN e, e.type as original_type, labels(e)[0] as sanitized_type;
9
10  MATCH (e:Entity)
11  WHERE e.type = $entity_type
12  RETURN e.name as name, e as node, e.id as framework_id;
13
14  MATCH (e:Entity {name: $entity_name})-[r]-(related:Entity)
15  WHERE e.type = $entity_type
16  RETURN r.rel_type as relationship_type,
17  related as target_entity,
18  r as relationship_properties,
19  type(r) as relation_class;
```

**Coverage & Entropy Metrics (reward proxies).** For retrieval coverage and structural diversity, we provide quantities computable within Neo4j: (i) coverage gain (proportion of newly introduced target-domain entities/relations), (ii) degree-distribution entropy (Shannon entropy as structural diversity proxy).

```
1   MATCH (n:Entity) WHERE n.source = $doc_id SET n.episode_tag = $ep;
2   MATCH ()-[r:REL]-() WHERE r.source = $doc_id SET r.episode_tag = $ep;
3   MATCH ()-[r:RELATIONSHIP]-() WHERE r.source = $doc_id SET r.episode_tag = $ep;
4
5   MATCH (n:Entity) WHERE n.episode_tag = $ep
6   WITH count(n) AS curE
7   MATCH (n:Entity) WHERE n.episode_tag = $prev_ep
8   WITH curE, count(n) AS prevE
9   RETURN (toFloat(curE - prevE) / greatest(1.0, prevE)) AS coverage_gain;
10
11  MATCH ()-[r:REL]-() WHERE r.episode_tag = $ep
12  WITH count(r) AS curR_framework
13  MATCH ()-[r:RELATIONSHIP]-() WHERE r.episode_tag = $ep
14  WITH curR_framework, count(r) AS curR_tool
15  MATCH ()-[r:REL]-() WHERE r.episode_tag = $prev_ep
16  WITH curR_framework, curR_tool, count(r) AS prevR_framework
17  MATCH ()-[r:RELATIONSHIP]-() WHERE r.episode_tag = $prev_ep
18  WITH curR_framework, curR_tool, prevR_framework, count(r) AS prevR_tool
```

```
19   WITH (curR_framework + curR_tool) AS curR, (prevR_framework + prevR_tool) AS prevR
20   RETURN (toFloat(curR - prevR) / greatest(1.0, prevR)) AS rel_coverage_gain;
21
22   MATCH (n:Entity)
23   OPTIONAL MATCH (n)-[r:REL]-()
24   OPTIONAL MATCH (n)-[rt:RELATIONSHIP]-()
25   WITH n, count(r) + count(rt) AS deg
26   WITH collect(deg) AS degs
27   UNWIND degs AS d
28   WITH d, size(degs) AS N, reduce(s=0, x IN degs | s + x) AS total_deg
29   WITH d, N, toFloat(d)/toFloat(total_deg) AS p
30   WHERE p > 0
31   WITH collect(-p * log(p)) AS terms
32   RETURN reduce(s=0.0, x IN terms | s + x) AS degree_entropy;
```

**Time-Consistency Regularizer.** Using normalized Laplacian $L = I - D^{-1/2}AD^{-1/2}$ with degree matrix $D$ and adjacency $A$. Neo4j exports sparse adjacency with stable node ordering for computing $\|L_{t+1} - L_t\|_F^2$ (for $R_{\text{env}}$ regularization term).

```
1    MATCH (n:Entity)
2    WITH n ORDER BY coalesce(n.id, n.name) ASC
3    WITH collect(coalesce(n.id, n.name)) AS ids_t
4    MATCH (s:Entity)-[r]->(o:Entity)
5    WHERE (r.episode_tag = $t OR r:RELATIONSHIP)
6    AND (r:REL OR r:RELATIONSHIP)
7    WITH ids_t, collect({
8      i: coalesce(s.id, s.name),
9      j: coalesce(o.id, o.name),
10     type: coalesce(r.rel_type, r.type)
11   }) AS edges_t
12   RETURN ids_t AS ids, edges_t AS edges;
13
14   MATCH (n:Entity)
15   WITH n ORDER BY coalesce(n.id, n.name) ASC
16   WITH collect(coalesce(n.id, n.name)) AS ids_tp1
17   MATCH (s:Entity)-[r]->(o:Entity)
18   WHERE (r.episode_tag = $tp1 OR r:RELATIONSHIP)
19   AND (r:REL OR r:RELATIONSHIP)
20   WITH ids_tp1, collect({
21     i: coalesce(s.id, s.name),
22     j: coalesce(o.id, o.name),
23     type: coalesce(r.rel_type, r.type)
24   }) AS edges_tp1
25   RETURN ids_tp1 AS ids, edges_tp1 AS edges;
```

**Tool Workflow Integration.** Queries supporting the six-tool Agentic-KGR workflow with unified access to both framework and tool-stored data.

```
1    MATCH (e:Entity)
2    WITH e.type as entity_type, count(e) as entity_count
3    MATCH ()-[r:REL]-()
4    WITH entity_type, entity_count, r.rel_type as rel_type, count(r) as rel_count
5    MATCH ()-[rt:RELATIONSHIP]-()
6    WITH entity_type, entity_count, rel_type, rel_count,
7    rt.type as tool_rel_type, count(rt) as tool_rel_count
8    RETURN entity_type, entity_count,
9    collect({type: rel_type, count: rel_count, source: "framework"}) +
10   collect({type: tool_rel_type, count: tool_rel_count, source: "tool"}) as relations;
11
12   WITH $schema_entity_types as expected_types
13   MATCH (e:Entity)
14   WITH expected_types, collect(distinct e.type) as found_types
```

```
15    RETURN [t IN expected_types WHERE NOT t IN found_types] as missing_types;
16
17    MATCH ()-[r:REL]-()
18    OPTIONAL MATCH (subj:Entity {id: r.src_id})
19    OPTIONAL MATCH (obj:Entity {id: r.dst_id})
20    WITH r, subj, obj, "framework" as source,
21    CASE WHEN subj IS NULL THEN 1 ELSE 0 END as missing_subject,
22    CASE WHEN obj IS NULL THEN 1 ELSE 0 END as missing_object
23
24    UNION ALL
25
26    MATCH ()-[rt:RELATIONSHIP]-()
27    OPTIONAL MATCH (subj_t:Entity {name: rt.subject})
28    OPTIONAL MATCH (obj_t:Entity {name: rt.object})
29    WITH rt as r, subj_t as subj, obj_t as obj, "tool" as source,
30    CASE WHEN subj_t IS NULL THEN 1 ELSE 0 END as missing_subject,
31    CASE WHEN obj_t IS NULL THEN 1 ELSE 0 END as missing_object
32
33    RETURN source, sum(missing_subject) as orphaned_subjects,
34    sum(missing_object) as orphaned_objects,
35    count(r) as total_relations;
```

**Snapshotting and Safety Checks.** For reproducibility and rollback, snapshot nodes record version boundaries. Safety checks enforce lightweight conflict rules before insertion.

```
1     CREATE (ss:GraphSnapshot {
2       sid: $sid,
3       created_at: datetime(),
4       note: $note,
5       tool_version: "agentic_kgr_v1"
6     });
7
8     MATCH (n:Entity) WHERE n.episode_tag = $ep OR n.source = $doc_id
9     SET n.snapshot = $sid;
10    MATCH ()-[r:REL]-() WHERE r.episode_tag = $ep SET r.snapshot = $sid;
11    MATCH ()-[r:RELATIONSHIP]-() WHERE r.source = $doc_id SET r.snapshot = $sid;
12
13    MATCH (s:Entity)-[r]-(o:Entity)
14    WHERE (r.episode_tag = $ep OR r.source = $doc_id)
15    AND (coalesce(s.id, s.name) = coalesce(o.id, o.name))
16    AND NOT coalesce(r.rel_type, r.type) IN $selfloop_whitelist
17    RETURN count(r) AS self_loops, collect(distinct coalesce(r.rel_type, r.type)) as
          types;
18
19    MATCH ()-[r]-()
20    WHERE (r.episode_tag = $ep OR r.source = $doc_id)
21    AND NOT coalesce(r.rel_type, r.type) IN $allowed_rel_types
22    RETURN coalesce(r.rel_type, r.type) AS invalid_type, count(*) AS n LIMIT 20;
```

**End-to-End Episode Pipeline.** Complete pipeline for one episode: extraction → staging → promotion → aging/cleanup → metrics evaluation → snapshotting, integrating both framework and tool operations.

```
1     :PARAM ep => 137, doc_id => 'wireless-42', tau_conf => 0.72, tau_votes => 3,
2     soft_window => 7, hard_window => 45, decay_rate => 0.08;
3
4     // 1) Stage framework candidates
5     UNWIND $candidates AS c
6     MERGE (s:Entity {id: c.src.id})
7     ON CREATE SET s.type = c.src.type, s.name = c.src.name
8     SET s.last_seen = datetime()
9     MERGE (o:Entity {id: c.dst.id})
10    ON CREATE SET o.type = c.dst.type, o.name = c.dst.name
```

```
11   SET o.last_seen = datetime()
12   MERGE (s)-[p:PENDING_REL {src_id:c.src.id, dst_id:c.dst.id, rel_type:c.rel.type}]->(o
     )
13   ON CREATE SET p.created_at = datetime(), p.confidence = c.rel.conf,
14   p.votes = 1, p.sources = [c.doc_id], p.episode_tag = $ep
15   ON MATCH SET p.votes = p.votes + 1, p.confidence = greatest(p.confidence, c.rel.conf)
     ;
16
17   // 2) Store tool entities and relationships
18   CALL apoc.periodic.iterate(
19   "UNWIND $tool_entities as entity RETURN entity",
20   "MERGE (e:Entity {name: entity.name, type: entity.type})
21   ON CREATE SET e.created_at = datetime(), e.source = 'agentic_kgr_tool'
22   SET e.last_seen = datetime()",
23   {batchSize: 1000, params: {tool_entities: $tool_entities}}
24   );
25
26   // 3) Promote staging to permanent relations
27   MATCH (s)-[p:PENDING_REL]->(o)
28   WHERE p.episode_tag = $ep AND p.confidence >= $tau_conf AND p.votes >= $tau_votes
29   MERGE (s)-[r:REL {src_id:p.src_id, dst_id:p.dst_id, rel_type:p.rel_type}]->(o)
30   ON CREATE SET r.created_at = datetime(), r.confidence = p.confidence,
31   r.evidence = p.sources, r.episode_tag = $ep
32   ON MATCH SET r.confidence = greatest(r.confidence, p.confidence),
33   r.evidence = apoc.coll.toSet(r.evidence + p.sources)
34   DELETE p;
35
36   // 4) Apply aging across all relation types
37   MATCH ()-[r]-()
38   WHERE r:REL OR r:RELATIONSHIP
39   WITH r, duration.between(r.last_seen, datetime()).days AS days
40   FOREACH (_ IN CASE WHEN days > $soft_window AND days <= $hard_window THEN [1] ELSE []
         END |
41   SET r.confidence = r.confidence * exp(-$decay_rate * (days - $soft_window))
42   )
43   FOREACH (_ IN CASE WHEN days > $hard_window THEN [1] ELSE [] END | DELETE r);
44
45   // 5) Create snapshot
46   CREATE (ss:GraphSnapshot {sid: toString($ep), created_at: datetime(),
47     note: 'unified-framework-tool-snapshot'});
48   MATCH (n:Entity) WHERE n.episode_tag = $ep OR n.source CONTAINS 'agentic_kgr'
49   SET n.snapshot = toString($ep);
50   MATCH ()-[r]-() WHERE r.episode_tag = $ep OR r.source CONTAINS 'agentic_kgr'
51   SET r.snapshot = toString($ep);
```

**Disambiguation Queries.** Supporting both exact match and semantic similarity strategies from the disambiguation tool. The exact match strategy provides deterministic results, while semantic similarity enables fuzzy matching.

```
1    MATCH (e {name: $entity_name})
2    WHERE labels(e)[0] = $entity_type
3    RETURN e, labels(e)[0] as type;
4
5    MATCH (e)
6    WHERE labels(e)[0] = $entity_type
7    AND toLower(e.name) = toLower($entity_name)
8    RETURN e, labels(e)[0] as type;
9
10   MATCH (e)
11   WHERE labels(e)[0] = $entity_type
12   RETURN e.name as name, e as node;
13
14   MATCH (e {name: $entity_name})-[r:RELATIONSHIP]-(related)
15   WHERE labels(e)[0] = $entity_type
16   RETURN r.type as relationship_type,
```

```
17  related as target_entity,
18  r as relationship_properties;
```

**Integration with Agentic-KGR Tool Workflow.**    The storage operations integrate with the six-tool workflow, supporting density assessment feedback, coverage analysis, and quality metrics through efficient queries.

```
1   MATCH (e)
2   UNWIND labels(e) as label
3   WITH label, count(e) as entity_count
4   MATCH ()-[r:RELATIONSHIP]->()
5   WITH label, entity_count, r.type as rel_type, count(r) as rel_count
6   RETURN label as entity_type, entity_count,
7   collect({type: rel_type, count: rel_count}) as relations;
8
9   WITH $schema_entity_types as expected_types
10  MATCH (e)
11  WITH expected_types, collect(distinct labels(e)[0]) as found_types
12  RETURN [t IN expected_types WHERE NOT t IN found_types] as missing_types;
13
14  MATCH ()-[r:RELATIONSHIP]->()
15  OPTIONAL MATCH (subj {name: r.subject})
16  OPTIONAL MATCH (obj {name: r.object})
17  WITH r, subj, obj,
18  CASE WHEN subj IS NULL THEN 1 ELSE 0 END as missing_subject,
19  CASE WHEN obj IS NULL THEN 1 ELSE 0 END as missing_object
20  RETURN sum(missing_subject) as orphaned_subjects,
21  sum(missing_object) as orphaned_objects,
22  count(r) as total_relations;
```

**Aging & Consistency.**    Relations not re-observed within window $\xi$ days are decayed or removed, corresponding to the aging and consistency penalty terms. This complements the tool's quality assessment capabilities.

```
1   MATCH ()-[r:RELATIONSHIP]->()
2   WHERE r.last_seen < datetime() - duration({days: $aging_threshold})
3   DELETE r;
4
5   MATCH ()-[r:RELATIONSHIP]->()
6   WHERE r.last_seen < datetime() - duration({days: $decay_threshold})
7   SET r.confidence = r.confidence * $decay_factor;
```

**Performance Optimizations.**    Optimizations specific to the tool implementation patterns, including batch processing and connection management.

```
1   UNWIND $entity_names as name
2   OPTIONAL MATCH (e {name: name})
3   WHERE labels(e)[0] = $entity_type
4   RETURN name, e IS NOT NULL as exists;
5
6   UNWIND $relations_batch as rel
7   OPTIONAL MATCH ()-[r:RELATIONSHIP {
8     type: rel.relation_type,
9     subject: rel.subject,
10    object: rel.object
11  }]->()
12  WITH rel, r IS NOT NULL as exists
13  WHERE NOT exists
14  RETURN rel;
15
16  CALL dbms.listConnections()
17  YIELD connectionId, connectTime, connector, userAgent
18  WHERE connectTime > datetime() - duration({minutes: 5})
```

```
19  RETURN count(*) as active_connections,
20  avg(duration.inSeconds(datetime(), connectTime)) as avg_duration;
```

**Error Handling and Validation.** Safety checks aligned with tool implementation error handling, including connection failures and data validation.

```
1   MATCH (e {name: $entity_name})
2   WITH e, labels(e) as current_labels, $expected_type as expected
3   WHERE NOT expected IN current_labels
4   RETURN e.name as inconsistent_entity,
5   current_labels, expected;
6
7   MATCH ()-[r:RELATIONSHIP]->()
8   OPTIONAL MATCH (subj {name: r.subject})
9   OPTIONAL MATCH (obj {name: r.object})
10  WHERE subj IS NULL OR obj IS NULL
11  RETURN r.type as problematic_relation,
12  r.subject, r.object,
13  subj IS NULL as missing_subject,
14  obj IS NULL as missing_object;
```

**Tool-Specific Metrics.** Metrics collection supporting the iterative feedback and quality assessment tools.

```
1   MATCH (e)
2   WITH labels(e)[0] as entity_type, count(e) as count
3   MATCH ()-[r:RELATIONSHIP]->()
4   WITH entity_type, count, r.type as rel_type, count(r) as rel_count
5   RETURN {
6     timestamp: datetime(),
7     entity_counts: collect({type: entity_type, count: count}),
8     relation_counts: collect({type: rel_type, count: rel_count}),
9     total_entities: sum(count),
10    total_relations: sum(rel_count)
11  } as iteration_snapshot;
12
13  WITH $schema_types as expected
14  MATCH (e)
15  WITH expected, collect(distinct labels(e)[0]) as found
16  UNWIND expected.entity_schema as expected_entity_type
17  WITH expected_entity_type, expected_entity_type IN found as covered
18  RETURN {
19    entity_coverage: {
20      total: size(expected.entity_schema),
21      covered: sum(CASE WHEN covered THEN 1 ELSE 0 END),
22      missing: [t IN expected.entity_schema WHERE NOT t IN found]
23    }
24  } as coverage_report;
```