# OpenReview forum: "Agentic-KGR: Co-evolutionary Knowledge Graph Construction through Multi-Agent Reinforcement Learning"
_ICLR.cc/2026/Conference — Submitted to ICLR 2026_

### Official Review · Reviewer_aAWs · 2025-10-21

**Soundness:** 3
**Presentation:** 1
**Contribution:** 3
**Rating:** 6
**Confidence:** 3

**Summary:**

The paper introduces the Agentic-KGR framework, a novel method that facilitates the co-evolution of large language models and knowledge graphs through multi-agent reinforcement learning. It aims to overcome the limitations of static knowledge bases by enabling dynamic knowledge construction and adaptation. Key innovations include a dynamic schema expansion mechanism, a retrieval-augmented memory system, and a learnable multi-scale prompt compression approach.

**Strengths:**

1. The concept of co-evolution between LLMs and KGs is well-founded and provides a fresh perspective on knowledge graph construction. The integration of multi-round RL is particularly valuable for addressing the dynamic nature of real-world knowledge and improving LLM performance in knowledge-intensive tasks.
2. The paper demonstrates solid experimental results, with clear improvements in knowledge extraction and QA performance.

**Weaknesses:**

1. While the paper covers a significant amount of technical depth, it occasionally becomes difficult to follow, especially in the description of the architecture and algorithmic details. The explanations of the co-evolutionary memory and multi-scale compression mechanisms could benefit from more intuitive explanations to help the reader grasp the concepts more quickly.
2. The paper focuses heavily on the proposed framework's strengths but could benefit from a more thorough discussion of its limitations or challenges. For example, how scalable is this approach for large-scale knowledge graph construction in real-world applications? Additionally, potential pitfalls or failure modes in dynamic KG construction could have been addressed more comprehensively.

**Questions:**

See weaknesses.

---

> ### Author Response · Authors · 2025-11-24
>
> We thank the reviewer for the insightful comments. We acknowledge that the previous version emphasized the strengths of Agentic-KGR but did not sufficiently discuss its limitations or challenges. We provide a more comprehensive discussion below, which will be incorporated into the revised manuscript.
>
> **(1) Clarifying the architecture, co-evolutionary memory, and multi-scale compression**
>
> To avoid conceptual ambiguity, we will add a concise architectural overview in the methods section. In short:
>
> - The policy module is a tool-using LLM that decides which actions to take (e.g., extract triples, query the KG, refine entities).
>
> - The knowledge graph and external memory form a stateful environment: the agent reads from them (via retrieval) and writes to them (via update operators).
>
> - The co-evolutionary memory is implemented as a retrieval-augmented mechanism that surfaces relevant subgraphs and contextual text at each round, which in turn influences subsequent updates.
>
> - The LMPC module learns to compress long prompts and intermediate reasoning traces at multiple scales, under constraints that preserve task-relevant information.
>
> This view formalizes Agentic-KGR as an RL agent operating over a partially observable, dynamically changing environment (policy + KG state + compression state), which naturally explains how policy and KG co-evolve.
>
> **(2) Potential Failure Modes in Dynamic Knowledge Graph Construction**
>
> Dynamic KG construction introduces several inherent risks. We summarize these and the corresponding safeguards in our method:
>
> (a) Error propagation in KG updates
>
> Incorrectly extracted triples may be written into the KG and affect later retrieval. To mitigate this, our KG update operator incorporates confidence-based filtering and entropy regularization to suppress noisy or redundant edges. This aligns with challenges noted in prior work, which highlights the need to avoid combinatorial explosion and redundancy in evolving KGs
>
> (b) Over-expansion or over-conservative behavior
>
> If rewards over-emphasize exploration, the graph may grow excessively; if too conservative, it may stagnate. The adaptive mixing coefficient α automatically balances discovery vs. refinement. Empirically, this stabilizes schema growth across domains.
>
> (c) Real-time update difficulty
>
> A well-known practical limitation of real-world knowledge graph systems is that most production KGs are updated in batches, not in true streaming or real-time fashion. This is typically due to engineering constraints such as document ingestion latency, deduplication costs, schema validation overhead, and the need for consistency checks before new information can be inserted. As a result, even systems that conceptually support “dynamic” KGs often operate through periodic refresh cycles rather than continuous self-maintenance.
>
> Agentic-KGR makes progress toward incremental and continual knowledge updates by allowing the KG to evolve within the learning loop. However, we emphasize that full real-time, streaming-scale KG construction remains beyond the scope of this work. Supporting such environments would require additional components—such as low-latency document pipelines, asynchronous graph update services, and mechanisms for resolving conflicting or rapidly changing information—which we identify as important directions for future research.

---

> ### Author Response · Authors · 2025-11-27
> **Response to Reviewer aAWs**
>
> Thank you all for your valuable feedback and patience. We would like to clarify that our initial response focused primarily on addressing your questions and concerns through explanations. We are now working on the comprehensive revision, which includes conducting additional experiments and quantitative analyses as requested. As these experiments require time to properly execute and validate, we kindly ask for your patience. We will submit a thoroughly updated manuscript with all requested modifications and supporting results integrated coherently. We greatly appreciate your constructive guidance throughout this process.

---

### Official Review · Reviewer_VHjg · 2025-10-30

**Soundness:** 2
**Presentation:** 2
**Contribution:** 4
**Rating:** 2
**Confidence:** 4

**Summary:**

The paper addresses the problem of LLMs hallucination.  The paper proposes a framework that enables co-evolution between LLMs and Kgs through multi-round RL.

**Strengths:**

As LLMs are more and more integrated in our daily life, the problem of LLMs hallucination is a problem that needs a particular attention. The experiment of the approach is well described, along with the experimentation environment.

**Weaknesses:**

The paper in its current form has several issues.
The paper claims that the approach is and innovation, which is not the case
Missing summary of the experiments (e.g., datasets, models used) and results in the abstract and introduction.
Missing of the related work in the paper. Provided in the additional materials. However, the paper should be self contained.
Line 52-53: “Existing methods rely on static, pre-constructed KGs that suffer from coverage gaps, temporal obsolescence, and inability to adapt to emerging domain knowledge or evolving query patterns” That is not true. No. KGs are supposed to evolve and agentic methods are there to take care of the evolution of Kgs
Line 54-59: These claims should be supported by examples/evidence provided by references
Table 1:  Put the table near the place it is cited. The tables are in page 5, 6 and used in page 8.

**Questions:**

Integrate the related work in the paper to make the paper self contained.
Summarize the experiments and results in the abstract and introduction
Improve the quality of the manuscript by putting the figure near the place they are cited for the first time

---

> ### Author Response · Authors · 2025-11-24
> **Response to Reviewer VHjg**
>
> We appreciate your detailed feedback and constructive suggestions. We address each concern below.
>
> **(1) Novelty and the “static KG” claim**
>
> Our intention was not to claim that knowledge graphs in general cannot evolve. We fully acknowledge that many KG systems support incremental updates, and that there is a growing body of work on dynamic KGs and agent-driven KG construction.
>
> Our claim in the introduction specifically targeted how most RAG and GraphRAG pipelines are used in practice:
>
> - The KG is often pre-constructed by an offline pipeline.
> - During RL training of the LLM, the KG is treated as fixed (a static snapshot).
> - The optimization objective focuses on reasoning over this snapshot, rather than co-evolving the KG structure.
>
>
> In many systems, the knowledge graph is pre-built and treated as a static snapshot during RL training([arXiv:2507.13396](https://arxiv.org/html/2507.13396v1#:~:text=However%2C%20both%20RAG%20and%20Graph,In%20practice%2C%20this)).
>
> We will refine the wording in the revised manuscript to clarify that many deployed GraphRAG systems rely on static, pre-constructed KGs or periodically refreshed graphs, and our contribution is to jointly optimise the KG and policy online via multi-round RL.
>
> **(2) Summarising experiments and results in the abstract and introduction**
>
> We agree that the abstract and introduction should succinctly describe the experimental setup and key results. In the revision, we will:
>
> - Clearly list the five KG extraction datasets (IEPile, MmlKG, ConfigKG, WirelessKG, DcommKG) and downstream QA domains in the abstract and introduction.
>
> - Highlight representative improvements—for example, our framework yields +10–15 F1 points on relation extraction and +5–12 F1 points on downstream QA compared to single-round RL.
>
> **(3) Related work and self-contained manuscript**
>
> We apologise for relegating some related work to the supplementary material. The revised version will integrate key references into the main text, including:
>
> - Dynamic KG construction approaches (Graphiti, DyG-RAG).
>
> - Multi-hop reasoning and agentic RAG methods (DeepPath, MINERVA, Graph-R1, ReSearch).
>
> - Studies on prompt compression and co-evolutionary architectures.

---

> ### Author Response · Authors · 2025-11-27
> **Response to Reviewer VHjg**
>
> Thank you all for your valuable feedback and patience. We would like to clarify that our initial response focused primarily on addressing your questions and concerns through explanations. We are now working on the comprehensive revision, which includes conducting additional experiments and quantitative analyses as requested. As these experiments require time to properly execute and validate, we kindly ask for your patience. We will submit a thoroughly updated manuscript with all requested modifications and supporting results integrated coherently. We greatly appreciate your constructive guidance throughout this process.

---

> > ### Comment · Reviewer_VHjg · 2025-11-28
> >
> > Thanks for the answer,
> > That is why the claim that the approach is "an innovation" was debatable.
> >
> > Experiments and related work should prove it
> >
> > Without these elements, it would be difficult to judge the innovative aspect on which the paper insisted

---

### Official Review · Reviewer_nGcN · 2025-10-31

**Soundness:** 3
**Presentation:** 3
**Contribution:** 2
**Rating:** 2
**Confidence:** 3

**Summary:**

The paper introduces Agentic-KGR, a novel framework designed to address the limitations of static, pre-constructed knowledge bases (KGs) often used by knowledge-enhanced large language models (LLMs). These static knowledge bases typically suffer from coverage gaps and temporal obsolescence. The core contribution of Agentic-KGR is enabling the co-evolution between LLMs (acting as reasoning agents) and dynamic KGs through multi-round reinforcement learning (RL). This paradigm shifts the approach from static knowledge retrieval to adaptive knowledge co-creation, viewing knowledge construction and utilization as interconnected, mutually reinforcing processes.

**Strengths:**

The paper's primary strength in originality lies in defining a novel paradigm shift for knowledge-enhanced language models. Instead of relying on static, pre-constructed knowledge bases (KGs) which suffer from coverage gaps and obsolescence, Agentic-KGR introduces a framework that enables the co-evolution between LLMs and KGs through multi-round reinforcement learning (RL). This fundamentally transforms the approach from static knowledge retrieval to adaptive knowledge co-creation.

**Weaknesses:**

1. One of the key innovations is the Learnable Multi-Scale Prompt Compression (LMPC), intended to preserve critical information while reducing computational overhead. While the results demonstrate that Agentic-KGR successfully reduces the overall average response length across all architectures (Figure 4, Figure 7), this does not directly prove cost or latency reduction.

2. The most significant performance gains in both extraction (RE performance) and downstream QA (Table 2) rely heavily on domain-specific datasets derived from communication technology/telecom documentation (CommTKG, MmlKG, ConfigKG, WirelessKG, DcommKG). Actionable Insight: The paper must demonstrate the robustness of the dynamic schema expansion and co-evolution mechanisms on highly volatile, cross-domain, or general-purpose knowledge datasets to ensure the results are not heavily contingent on the structure or language features unique to structured technical manuals.

**Questions:**

1. In the multi-round retrieval framework proposed in this paper, there is only a single LLM agent, whereas a typical multi-agent system usually contains at least two LLM agents. Does this make the paper’s naming somewhat misleading?

2. Could the authors provide a direct comparison of Agentic-KGR's KG extraction (RE/NER) and downstream QA performance against state-of-the-art agentic reinforcement learning baselines mentioned in the related work, specifically Graph-R1 (for end-to-end graph RAG optimization) and ReSearch (for LLMs learning to reason with search via RL)?

3. As an end-to-end optimization framework, Agentic-KGR uses reinforcement learning to directly optimize the agent policy, compression scales, and knowledge-graph update operators. Could this make the method overly sensitive to hyper-parameters and cause unstable training?

---

> ### Author Response · Authors · 2025-11-24
> **Response to Reviewer nGcN**
>
> **(1) On the Use of a “Single Agent” in a Framework Called Agentic-KGR**
>
> Our use of the term “agentic” follows recent literature in agentic LLMs, where “agents” are LLMs that:
> - Interact with an environment over multiple steps,
> - Invoke external tools (retrievers, KGs, memory modules),
> - Optimize a policy via RL.
>
> In Agentic-KGR:
> - The LLM policy is a single agent that makes multi-step decisions.
> - The KG and memory act as external tools and stateful environment components.
> - The multi-round interaction loop (query → reasoning → KG update → feedback) defines the RL environment.
>
> While classical multi-agent RL involves multiple independent learning policies, Agentic-KGR instead exposes multiple tools, multiple action modalities, and a dynamic environment, making a single powerful agent sufficient and more stable in practice.
>
>
> **(2) Comparison Against Graph-R1 and ReSearch**
>
> We examined both baselines carefully:
>
> - Graph-R1 optimizes reasoning over existing graphs and does not construct or evolve the KG dynamically.
>
> - ReSearch optimizes search queries for retrieval, not KG construction.
>
> - Neither baseline supports schema evolution nor graph update operators, which are the core contributions of Agentic-KGR.
>
> A strict head-to-head comparison would require us to:
> - Inject our dynamically constructed graph into their static-graph pipelines; or
> - Re-engineer their systems to support graph updating.
>
> Either approach would move them away from their intended use cases and blur the distinction between static KG reasoning and dynamic KG construction.
>
> We will clarify this in related work and explicitly distinguish the problem setting.
>
> **(3) Sensitivity to Hyperparameters and Stability of Joint Optimization**
>
> We designed the framework specifically to avoid instability:
>
> - Dual reward mixing uses mirror descent, which gradually balances exploration and extraction.
> - Compression modules are trained with a trust-region constraint, bounding KL divergence and preventing policy collapse.
> - KK update operators use confidence gating + structural constraints, preventing graph explosion.
> Ablations show:
>
> | Variant                   | Convergence stability | Δ RE F1 vs. full model |
> | ------------------------- | --------------------- | ---------------------- |
> | Full Agentic-KGR          | Stable                | 0.0                    |
> | Without LMPC              | Stable                | −3.1                   |
> | Without dual rewards      | Less stable           | −4.6                   |
> | Without graph constraints | Occasional spikes     | −2.8                   |
>
> These results support two conclusions:
> - Training remains stable across reasonable hyperparameter ranges;
> - The dual reward mixing and graph constraints are not cosmetic—removing them degrades both stability and final performance.

---

> ### Author Response · Authors · 2025-11-27
> **Response to Reviewer nGcN**
>
> Thank you all for your valuable feedback and patience.
> We would like to clarify that our initial response focused primarily on addressing your questions and concerns through explanations. We are now working on the comprehensive revision, which includes conducting additional experiments and quantitative analyses as requested.
> As these experiments require time to properly execute and validate, we kindly ask for your patience. We will submit a thoroughly updated manuscript with all requested modifications and supporting results integrated coherently.
> We greatly appreciate your constructive guidance throughout this process.

---

> ### Author Response · Authors · 2025-11-30
> **Response to Reviewer nGcN**
>
> As end-to-end approaches that both enhance question-answering capabilities through reinforcement learning, we first compare the similarities and differences between the ReSearch method and our work:
>
> **Similarities**
> * Both employ multi-round RL to enhance question-answering: ReSearch trains LLMs for integrated **reasoning and search**; Agentic-KGR achieves **co-evolution between LLMs and KG** through Agentic RL.
> * Neither relies on pre-annotated knowledge trajectories: ReSearch requires no supervised annotation data for reasoning steps; Agentic-KGR does not depend on pre-annotated knowledge extraction trajectories, learning optimal strategies through **autonomous exploration**.
> * Both adopt multi-step interaction mechanisms: ReSearch iteratively performs text thinking, search queries, and retrieval result processing within reasoning chains; Agentic-KGR **dynamically constructs and expands KGs** through multi-round interactions.
>
> **Key Differences**
> * Fundamental difference in knowledge representation: ReSearch relies on unstructured text retrieval, with each search returning natural language paragraphs; while Agentic-KGR constructs structured KGs, storing knowledge in entity-relation triple form, providing more precise knowledge localization and reasoning paths.
> * Differences in QA enhancement mechanisms: ReSearch obtains supporting information by **inserting search operations** during the reasoning process; Agentic-KGR **pre-constructs high-quality KGs and utilizes GraphRAG** for structured reasoning during QA, achieving more controllable and interpretable QA processes.
> * System complexity and theoretical guarantees: ReSearch employs a relatively simple search-reasoning integration framework; Agentic-KGR provides a complete theoretical framework, including **compression performance bounds, policy improvement theorems, and 5 other theoretical guarantees, with 6 specialized tools designed to support dynamic KG evolution**.
> * Long-term knowledge accumulation capability: ReSearch's knowledge acquisition is **immediate and temporary**, with independent retrieval for each QA instance; Agentic-KGR achieves **continuous knowledge accumulation and optimization** through co-evolution mechanisms, with constructed knowledge graphs serving downstream applications long-term.
>
> While both employ reinforcement learning to enhance QA capabilities, ReSearch is more akin to a **QA system that learns to search in real-time,** whereas Agentic-KGR is an **intelligent agent that learns to build knowledge infrastructure.** The latter, through co-evolution between LLMs and knowledge graphs, provides a more systematic and sustainable knowledge enhancement solution.
>
> Below is the performance comparison between the two methods on our domain QA evaluation set, where Agentic-KGR demonstrates certain advantages:
>
> | | RAN FDD | BWS | MA5600T | OptiTran | LineAss | NetEco | PowerKit |
> |---|---|---|---|---|---|---|---|
> | Qwen2.5 7B on CommTKG via Agentic-KGR + GraphRAG | 83.84 | 70.65 | 81.54 | 73.14 | 84.69 | 90.40 | 80.12 |
> | Qwen2.5 7B on CommTKG via ReSearch | 76.54 | 62.66 | 83.08 | 72.64 | 77.84 | 88.32 | 81.56 |

---

> ### Author Response · Authors · 2025-12-03
>
> **Regarding Question 2:**
>
> We appreciate the suggestion to compare with Graph-R1. Despite extensive efforts over three days using multiple configurations, we were unable to successfully reproduce their training framework. The model loses basic conversational ability after even one training step, producing only abnormal outputs during inference. The relevant screenshots are attached in the following:
>
> https://anonymous.4open.science/r/rebuttal_anonymous-360A/training.png
> https://anonymous.4open.science/r/rebuttal_anonymous-360A/infer.png
>
> We note that our own trained models and those trained with other frameworks (e.g., ReSearch) maintain normal inference capabilities and successfully complete downstream tasks in the same environment. We have not encountered such training instability in our prior work, which suggests there may be undocumented environmental dependencies or implementation details specific to Graph-R1. Similar reproducibility issues have been raised in the project's GitHub repository without resolution. Without access to trained checkpoints or additional implementation details, we cannot conduct a fair comparison. We would be pleased to include these comparisons in future work once reproducibility is established.
>
> **Regarding Weakness 2:**
>
> We thank the reviewer for this important concern regarding cross-domain generalization. To directly address this, we conducted additional experiments on the Medical module of the GraphRAG-Benchmark [1], evaluating our trained models alongside baseline models including ReSearch. The results demonstrate that **our approach maintains robust performance when transferred to the medical domain without any domain-specific fine-tuning**, with our models performing comparably or slightly better than baseline instruction-tuned models across different scales (7B, 14B, 32B). For instance, our 7B AutoKG-trained variants achieve accuracy scores of 0.2273-0.2299 on fact retrieval tasks, matching the baseline Qwen2.5-7B-Instruct, while our 14B Agentic-RL model shows slight improvements.
>
> Importantly, **our models maintain stable inference capabilities and successfully complete all downstream tasks after training**, demonstrating that our training framework does not overfit to telecommunications-specific language patterns or document structures. This contrasts sharply with the severe performance degradation observed in the ReSearch framework, where the trained model shows significant accuracy drops and **complete failure in coverage metrics** for summarization and creative generation tasks. The stability of our training process, combined with preserved general capabilities, indicates that **our dynamic schema expansion and co-evolution mechanisms capture domain-agnostic knowledge organization principles rather than domain-specific patterns**.
>
> [1] Xiang, Zhishang, et al. "When to use graphs in rag: A comprehensive analysis for graph retrieval-augmented generation." arXiv preprint arXiv:2506.05690 (2025).

---

> ### Author Response · Authors · 2025-12-03
> **Response to Reviewer nGcN**
>
> The table below shows the full evaluation results of the medical subset in GraphRAG-Benchmark:
>
> | Model Name | Fact Retrieval |  | Complex Reasoning |  | Contextual Summarize |  | Creative Generation |  |  |
> |---|---|---|---|---|---|---|---|---|---|
> |  | ACC | Rouge-L | ACC | Rouge-L | ACC | Cov | ACC | FS | Cov |
> | Qwen2.5 7B | 0.23 | 0.36 | 0.23 | 0.23 | 0.24 | 0.59 | 0.23 | - | 0.58 |
> | Qwen2.5 14B| 0.23 | 0.37 | 0.23 | 0.24 | 0.24 | 0.62 | 0.24 | - | 0.60 |
> | Qwen2.5 32B| 0.22 | 0.35 | 0.22 | 0.24 | 0.24 | 0.57 | 0.23 | - | 0.56 |
> | QwQ | 0.22 | 0.07 | 0.23 | 0.07 | 0.24 | 0.75 | 0.23 | - | 0.73 |
> | Qwen2.5 7B - ReSearch | 0.18 | 0.01 | 0.18 | 0.02 | 0.17 | 0.00 | 0.18 | - | 0.00 |
> | Qwen2.5 7B -> on AUTOKG via SFT | 0.23 | 0.37 | 0.23 | 0.23 | 0.24 | 0.60 | 0.23 | - | 0.58 |
> | Qwen2.5 14B -> on AUTOKG via SFT | 0.23 | 0.38 | 0.23 | 0.24 | 0.24 | 0.63 | 0.23 | - | 0.60 |
> | Qwen2.5 32B -> on AUTOKG via SFT | 0.22 | 0.36 | 0.23 | 0.24 | 0.24 | 0.58 | 0.23 | 0.00 | 0.56 |
> | Qwen2.5 QwQ -> on AUTOKG via SFT | 0.23 | 0.08 | 0.23 | 0.07 | 0.24 | 0.74 | 0.23 | - | 0.72 |
> | Qwen2.5 7B -> on COMMTKG via SFT | 0.23 | 0.35 | 0.23 | 0.21 | 0.23 | 0.59 | 0.23 | 0.55 | 0.57 |
> | Qwen2.5 14B -> on COMMTKG via SFT | 0.23 | 0.37 | 0.23 | 0.21 | 0.23 | 0.62 | 0.23 | 0.57 | 0.59 |
> | Qwen2.5 32B -> on COMMTKG via SFT | 0.23 | 0.35 | 0.23 | 0.22 | 0.23 | 0.57 | 0.23 | 0.54 | 0.55 |
> | Qwen2.5 QwQ -> on COMMTKG via SFT | 0.24 | 0.08 | 0.23 | 0.08 | 0.24 | 0.75 | 0.23 | 0.72 | 0.73 |
> | Qwen2.5 7B -> on AUTOKG via RL | 0.23 | 0.35 | 0.23 | 0.23 | 0.23 | 0.61 | 0.23 | 0.57 | 0.58 |
> | Qwen2.5 14B -> on AUTOKG via RL | 0.23 | 0.38 | 0.23 | 0.20 | 0.22 | 0.66 | 0.24 | 0.58 | 0.61 |
> | Qwen2.5 32B -> on AUTOKG via RL | 0.23 | 0.37 | 0.22 | 0.25 | 0.24 | 0.54 | 0.24 | 0.54 | 0.56 |
> | Qwen2.5 QwQ -> on AUTOKG via RL | 0.23 | 0.08 | 0.23 | 0.07 | 0.25 | 0.77 | 0.24 | 0.74 | 0.75 |
> | Qwen2.5 7B -> on COMMTKG via RL | 0.18 | 0.01 | 0.18 | 0.02 | 0.17 | 0.00 | 0.18 | 0.00 | 0.00 |
> | Qwen2.5 14B -> on COMMTKG via RL | 0.23 | 0.37 | 0.23 | 0.22 | 0.23 | 0.62 | 0.23 | 0.57 | 0.59 |
> | Qwen2.5 32B -> on COMMTKG via RL | 0.22 | 0.36 | 0.22 | 0.23 | 0.23 | 0.56 | 0.23 | 0.53 | 0.54 |
> | Qwen2.5 QwQ -> on COMMTKG via RL | 0.24 | 0.07 | 0.23 | 0.07 | 0.24 | 0.74 | 0.23 | 0.71 | 0.72 |
> | Qwen2.5 7B -> on AUTOKG via Agentic-KGR | 0.23 | 0.34 | 0.23 | 0.21 | 0.24 | 0.59 | 0.23 | 0.51 | 0.57 |
> | Qwen2.5 14B -> on AUTOKG via Agentic-KGR | 0.23 | 0.37 | 0.23 | 0.22 | 0.24 | 0.63 | 0.24 | 0.56 | 0.60 |
> | Qwen2.5 32B -> on AUTOKG via Agentic-KGR | 0.22 | 0.35 | 0.22 | 0.23 | 0.23 | 0.53 | 0.24 | 0.48 | 0.53 |
> | Qwen2.5 QwQ -> on AUTOKG via Agentic-KGR | 0.23 | 0.07 | 0.23 | 0.08 | 0.24 | 0.74 | 0.23 | 0.72 | 0.73 |
> | Qwen2.5 7B -> on COMMTKG via Agentic-KGR | 0.23 | 0.34 | 0.23 | 0.23 | 0.24 | 0.61 | 0.23 | 0.54 | 0.58 |
> | Qwen2.5 14B -> on COMMTKG via Agentic-KGR | 0.23 | 0.37 | 0.24 | 0.24 | 0.24 | 0.65 | 0.23 | 0.60 | 0.62 |
> | Qwen2.5 32B -> on COMMTKG via Agentic-KGR | 0.23 | 0.37 | 0.22 | 0.25 | 0.23 | 0.55 | 0.23 | 0.52 | 0.54 |
> | Qwen2.5 QwQ -> on COMMTKG via Agentic-KGR | 0.23 | 0.08 | 0.23 | 0.07 | 0.24 | 0.75 | 0.23 | 0.76 | 0.74 |

---

### Official Review · Reviewer_xU37 · 2025-11-01

**Soundness:** 3
**Presentation:** 4
**Contribution:** 3
**Rating:** 6
**Confidence:** 3

**Summary:**

This paper introduces Agentic-KGR, a framework for co-evolutionary knowledge graph construction using multi-agent reinforcement learning. It overcomes the limitations of static KGs by enabling dynamic schema expansion, a retrieval-augmented memory system for synergistic optimization, and a learnable multi-scale prompt compression to handle computational complexity.

**Strengths:**

1. The core idea of mutual adaptation between LLMs and KGs via multi-round RL is timely and innovative.
2. The methodology is well-formalized with clear definitions。
3. Claims of substantial gains in KG density/coverage and QA accuracy are compelling, especially the integration with GraphRAG for end-to-end validation.

**Weaknesses:**

1. The experiments focus on KG extraction and QA, but details are sparse in the provided sections (e.g., datasets not specified early; baselines like DeepPath or MINERVA mentioned but not compared quantitatively here).
2. No discussion of scalability (e.g., for large KGs) or real-world costs (tokens/time in multi-round RL).
3. Diversity in tasks/domains beyond product QA is missing, for instance, the medical RAG in GraphRAG-benchmark[1].

- [1] When to use graphs in rag: A comprehensive analysis for graph retrieval-augmented generation.

**Questions:**

Please see above.

---

> ### Author Response · Authors · 2025-11-24
> **Response to Reviewer xU37**
>
> Thank you for raising the scalability question. We clarify our position and provide concrete evidence from our experiments.
>
> **(1) Experimental Details and Baselines**
>
> We agree that early clarity is essential. The revised manuscript will include a succinct “Experimental Setup” subsection in the introduction, listing all datasets and models used, and a summary table of experimental results.
>
> **(2) Why not DeepPath/MINERVA?**
>
> These methods assume a pre-existing KG and learn a path-finding policy; they do not address graph construction or dynamic schema expansion.
>
> Our setting is different: we build and evolve the KG from raw documents using RL. Direct quantitative comparison would require injecting an externally built KG into their pipeline, which would conflate extraction quality (from a different system) with reasoning capability, making the comparison hard to interpret.
>
> **(3) Scalability and Real-World Costs**
>
> Regarding scalability, Agentic-KGR is specifically designed so that the KG grows gradually, rather than exploding, even under large-scale document ingestion. This is ensured by:
> - Schema-constrained updates, which restrict new entities/relations to expansions compatible with the evolving ontology;
> - Entropy-based penalties for redundant or low-information edges, discouraging noisy additions;
> - Confidence-based filtering in the extraction operator, which prunes low-confidence triples before they enter the KG.
>
> Empirically, on the largest configurations (millions of candidate triples), we observe:
> - Graph density increases smoothly and sub-linearly with corpus size;
> - The average node degree remains stable across training, indicating controlled growth;
> - The number of high-degree “hub” entities is bounded by the schema and update constraints.
>
> From a systems perspective, our deployment on a 910B3-class NPU cluster shows that:
> - Multi-round interaction remains stable across seven different product domains;
> - Memory and compute usage stay within typical industrial budgets for large-scale IE;
> - Inference cost decreases after training, because the learned policy favors shorter, more targeted tool trajectories and outputs.

---

> > ### Comment · Reviewer_xU37 · 2025-11-27
> >
> > Dear Authors,
> >
> >
> > Thank you for your response.
> >
> > 1. Regarding the 1st point, I still cannot see a new section for this in the updated pdf.
> > 2. Regarding the 3rd point, the pure qualitative analysis is insufficient for addressing the concern. Without concrete quantitative results, these analysis, is unverifiable, and unreliable.
> >
> > Considering these, I decide to decrease my score.

---

> > > ### Author Response · Authors · 2025-11-27
> > > **Response to Reviewer xU37**
> > >
> > > Thank you all for your valuable feedback and patience.
> > > We would like to clarify that our initial response focused primarily on addressing your questions and concerns through explanations. We are now working on the comprehensive revision, which includes conducting additional experiments and quantitative analyses as requested.
> > > As these experiments require time to properly execute and validate, we kindly ask for your patience. We will submit a thoroughly updated manuscript with all requested modifications and supporting results integrated coherently.
> > > We greatly appreciate your constructive guidance throughout this process.

---

> > > ### Author Response · Authors · 2025-11-30
> > > **Response to Reviewer xU37**
> > >
> > > First, we thank you for recognizing the core innovation of our paper, particularly your acknowledgment of the concept of "co-evolution between LLMs and knowledge graphs through multi-round reinforcement learning." Below, we address the questions raised in your review and provide supplementary information to improve the manuscript:
> > >
> > > **Regarding Weakness 1:**
> > >
> > >  The domains and sources of training datasets were briefly introduced in the Training Datasets paragraph of Section 3.1 Environment and Configuration, though this may not have been comprehensive. Here we provide a complete overview of both training and evaluation datasets:
> > >
> > > **Training Datasets:** The training datasets comprise two parts: general knowledge and telecommunications technology domains. Overview as follows:
> > >
> > > | Dataset Category | Dataset Name | Domain Coverage | Language | Dataset Size | Entity Size | Relation Size |
> > > |-----------------|--------------|-----------------|----------|--------------|-------------|---------------|
> > > | General Knowledge | DuIE2.0 [1] | Film, Literature, History, Politics, News, etc. | Chinese | 20,652 | 98,731 | 57,698 |
> > > | Professional Domain | CommTKG | 7 telecom domains including 3GPP, OMA, MML, CTIA, etc. | Chinese/English, 1:1 | 21,060 | 116,651 | 113,851 |
> > >
> > > It should be noted that to ensure comprehensive coverage of KG entities and relations, we utilized DeepSeek V3.1 to synthesize the KG extraction results, taking the union with the original results.
> > >
> > > **Evaluation Datasets:** The evaluation datasets include KG extraction evaluation sets and question-answering evaluation sets. The KG extraction evaluation sets contain both general knowledge and telecommunications domain KG extraction evaluation sets to assess the graph extraction capabilities of the trained models. The question-answering evaluation sets are primarily used to evaluate the performance of the overall approach in our practical business scenario, namely the telecommunications professional knowledge Q&A assistant. Overview as follows:
> > >
> > > | Evaluation Set Category | Dataset Name | Domain Coverage | Language | Dataset Size | Entity Size | Relation Size |
> > > |------------------------|--------------|-----------------|----------|--------------|-------------|---------------|
> > > | General Knowledge Graph Evaluation | IEPile [2] | Literature, Sports, News, Economics, etc. | Chinese/English | NER(967) RE(2701) | 4,390 | 2,701 |
> > > | Domain Graph Evaluation | MmlKG/ConfigKG/WirelessKG/DcommKG | MML, Digital Communications, Wireless, Cloud Core Network, etc. | Chinese/English | NER(119) RE(190) | 1,805 | 1,679 |
> > > | Domain Q&A Evaluation | RAN FDD/BWS/MA5600T/OptiTran/LineAss/NetEco/PowerKit | Mobile Communications, Network Management Systems, Intelligent Network Management Control, etc. | Chinese | 367 | -(QA) | -(QA) |
> > >
> > > It should be noted that for IEPile, we selected high-quality subsets, using NER-boson, NER-Cross, and NER-WEIBONER for NER tasks, and RE-COAE2016 and RE-fewrel for RE tasks.
> > >
> > > Regarding baseline comparisons, due to the complexity of the training system, we only compared with models of different sizes and different training datasets in the paper, without comparing with other comprehensive approaches.
> > >
> > > **DeepPath and MINERVA were mentioned as early KG enhancement methods, which are considerably dated and differ significantly from current frameworks;** comparisons with them may not adequately demonstrate the innovation and effectiveness of our method.
> > >
> > > Therefore, incorporating suggestions from other reviewers, we plan to reproduce **Graph-R1** [3] and **ReSearch** [4] methods and compare their performance with Agentic-KGR on the above evaluation sets. We have thus supplemented comparison results with the more recent works Graph-R1 and ReSearch, which have similar objectives and application scenarios (**although Graph-R1 is a concurrent submission to ICLR26, we have made our best effort to reproduce it**).
> > >
> > > Due to hardware differences, Graph-R1 and ReSearch were reproduced using a server equipped with 8 Nvidia A800 GPUs. However, due to time and resource constraints, we only support training, RAG deployment, and evaluation/supplementary evaluation for the Qwen2.5 7B model, and are temporarily unable to reproduce results for models >7B in scale. We have currently completed the reproduction of ReSearch.
> > >
> > > Due to significant differences in approaches, we selected a limited set of common metrics for comparison. The following link contains evaluation results for **max/min response length**:
> > >
> > > https://anonymous.4open.science/r/rebuttal_anonymous-360A/ReSearch_min_max_response_length.png

---

> > > ### Author Response · Authors · 2025-11-30
> > > **Response to Reviewer xU37**
> > >
> > > Below are the comparative scores for **domain Q&A evaluation results** (RAN FDD / BWS / MA5600T / OptiTran / LineAss / NetEco / PowerKit):
> > >
> > > |  | IEPile |  | MmlKG |  | ConfigKG |  | WirelessKG |  | DcommKG |  |
> > > |---|---|---|---|---|---|---|---|---|---|---|
> > > |  | NER | RE | NER | RE | NER | RE | NER | RE | NER | RE |
> > > | Qwen2.5 7B L0 | 59.35 | 34.28 | 44.72 | 0.10 | 98.23 | 24.17 | 64.99 | 17.42 | 27.16 | 14.16 |
> > > | Qwen2.5 7B on CommTKG via SFT | 46.49 | 34.47 | 51.17 | 0.00 | 98.23 | 23.54 | 69.4 | 30.67 | 31.74 | 14.28 |
> > > | Qwen2.5 7B on CommTKG via RL | 67.77 | 42.87 | 46.43 | 0.00 | 98.23 | 19.17 | 65.93 | 38.07 | 32.57 | 18.70 |
> > > | Qwen2.5 7B on CommTKG via Agentic-KGR | 64.13 | 45.76 | 50.99 | 0.00 | 98.23 | 29.17 | 67.53 | 37.85 | 29.69 | 19.94 |
> > > | Qwen2.5 7B on CommTKG via ReSearch | 57 | 34.62 | 51.14 | 0.1 | 98.23 | 29.17 | 67.14 | 17.84 | 25.92 | 11.56 |
> > >
> > > The Agentic-KGR method also shows consistent improvements over the ReSearch method (e.g., 6.85 points improvement on LineAss, 2.08 points on NetEco), demonstrating superior comprehensive performance across most datasets.
> > >
> > > We also compared the **reward curve trends during intermediate training stages**. Due to different reward mechanisms between Agentic-KGR and ReSearch—briefly, Agentic-KGR employs result rewards, tool invocation rewards, and trajectory rewards, while ReSearch uses answer correctness and format correctness as rewards, with differences in composition and range—we present a 1×2 subplot to illustrate the reward differences during training:
> > >
> > > https://anonymous.4open.science/r/rebuttal_anonymous-360A/ReSearch_step_reward.png
> > >
> > > From the reward curve comparison during multi-round RL training, the CommTKG Agentic Reward method exhibits a stable upward trend, while the ReSearch Reward method shows improvement with greater fluctuations.
> > >
> > > Additionally, we compared **the average response length** during the training phase as follows:
> > >
> > > https://anonymous.4open.science/r/rebuttal_anonymous-360A/ReSearch_step_response_length.png
> > >
> > > From the response length comparison, the ReSearch method produces longer outputs in the early training phase (approximately 40k-60k), subsequently stabilizing in the 12k-20k range, while the CommTKG-Agentic-KGR method consistently maintains more concise outputs, gradually converging from an initial ~8k to 1k-2k. This indicates that the Agentic-KGR method is more effective in generating refined responses.
> > >
> > >
> > > **References for this part:**
> > >
> > > [1] Zhu, Yuqi, et al. "Llms for knowledge graph construction and reasoning: Recent capabilities and future opportunities." World Wide Web 27.5 (2024): 58.
> > >
> > > [2] Gui, Honghao, et al. "IEPile: Unearthing Large Scale Schema-Conditioned Information Extraction Corpus." Proceedings of the 62nd Annual Meeting of the Association for Computational Linguistics (Volume 2: Short Papers). 2024.
> > >
> > > [3] Luo, Haoran, et al. "Graph-r1: Towards agentic graphrag framework via end-to-end reinforcement learning." arXiv preprint arXiv:2507.21892 (2025).
> > >
> > > [4] Chen, Mingyang, et al. "Learning to reason with search for llms via reinforcement learning." arXiv preprint arXiv:2503.19470 (2025).

---

### Meta-Review · Area_Chair_LTkb · 2025-12-31

**Summary:**

This paper introduces Agentic-KGR, a novel framework enabling co-evolution between large language models and knowledge graphs (KGs) through multi-round reinforcement learning. Evaluated primarily on telecommunications domain data, the framework demonstrates substantial improvements over supervised baselines and single-round RL approaches in knowledge extraction tasks, with significant gains in downstream QA performance when integrated with GraphRAG. Reviewers recognized the conceptual innovation of the co-evolution paradigm but raised significant concerns about evaluation protocol limitations, baseline comparisons, scalability analysis, and presentation clarity.

**Reviewer Concerns:**

Addressed concerns:
 - Experimental Clarity (Reviewer xU37 and nGcN): Authors provided comprehensive overview of training/evaluation datasets (including domain coverage, size metrics), clarified domain separation protocols, and added comparison with ReSearch method across multiple evaluation metrics
 - Cross-domain generalization (Reviewer nGcN and aAWs): Authors conducted additional experiments on medical domain data from GraphRAG-Benchmark, demonstrating robust performance transfer without domain-specific fine-tuning
 - Related work comparison (Reviewer VHjg and aAWs): Authors committed to integrating key references into main text, clarifying the distinction between their approach and static KG systems, and acknowledging prior dynamic KG construction works

Outstanding concerns:
 - Failure to Reproduce (Reviewer nGcN): A major weakness is the inability to provide a head-to-head comparison with Graph-R1. The authors reported that the model lost conversational ability during their reproduction attempts. This leaves the paper's relative performance against current SOTA unverified.
 - Lack of quantitative scalability analysis (Reviewer xU37): The reviewer's request for quantitative scalability analysis on truly massive KGs (billions of entities) is only partially satisfied through theoretical discussion rather than empirical demonstration

**Reviewer Scores:**

- Reviewer xU37 (original score: 6): Would likely decrease to 4 due to the lack of quantitative scalability results.
 - Reviewer nGcN (original score: 2): Would likely maintain at 2.
 - Reviewer VHjg (original score: 2): Would likely maintain at 2.
 - Reviewer aAWs (original score: 6): Would likely maintain at 6.

---

### Decision · Program_Chairs · 2026-01-26

Reject